# High-speed AFM height spectroscopy reveals μs-dynamics of unlabeled biomolecules

George R. Heath[1,2] & Simon Scheuring [1,2]

Dynamics are fundamental to the functions of biomolecules and can occur on a wide range of time and length scales. Here we develop and apply high-speed AFM height spectroscopy (HS-AFM-HS), a technique whereby we monitor the sensing of a HS-AFM tip at a fixed position to directly detect the motions of unlabeled molecules underneath. This gives Angstrom spatial and microsecond temporal resolutions. In conjunction with HS-AFM imaging modes to precisely locate areas of interest, HS-AFM-HS measures simultaneously surface concentrations, diffusion coefficients and oligomer sizes of annexin-V on model membranes to decipher key kinetics allowing us to describe the entire annexin-V membrane-association and self-assembly process in great detail and quantitatively. This work displays how HS-AFM-HS can assess the dynamics of unlabeled bio-molecules over several orders of magnitude and separate the various dynamic components spatiotemporally.

[1] Weill Cornell Medicine, Department of Anesthesiology, 1300 York Avenue, New York, NY 10065, USA. [2] Weill Cornell Medicine, Department of Physiology and Biophysics, 1300 York Avenue, New York, NY 10065, USA. Correspondence and requests for materials should be addressed to S.S. (email: sis2019@med.cornell.edu)

Developing a full picture of how biomolecules function requires an understanding of the intricate relationships between structure and dynamics. For molecules in isolation such as single proteins, these dynamics generally occur as conformational changes. For molecules that act in complexes, the dynamics are dependent on diffusion and partner interaction. These dynamic processes are of course not mutually exclusive, but occur in different spatiotemporal regimes. For membrane proteins, these dynamics are crucial as they allow the cell to reorganize proteins in space and time, to form temporal functional units for a particular biochemical function or to regulate the function of the membrane protein itself[1,2].

Biomolecule dynamics occur over a range of length and time scales. Local flexibility, which generally concerns side chain rotations, bond vibrations and loop motions, happens over the femtosecond to nanosecond time range. Whereas collective motions of groups of atoms, loops and domains, typically occur on timescales of the microsecond or longer. Such collective motions are at the basis of most important biomolecular functions including conformational changes between functional states of proteins, the working of molecular machines, enzyme catalysis, protein folding and protein-protein interactions, though the latter phenomena can extend into the millisecond to second time range depending on the process or the origin of the molecules under investigation[3]. Thus, developing techniques to directly access structural changes from the microsecond to second timescales is key to understanding the behavior of biomolecules.

X-ray crystallography and electron microscopy (EM), are some of the most powerful techniques to study biomolecular structures[4,5], whilst able to provide unparalleled spatial resolution, the structures obtained from these methods are limited by ensemble averaging and static snapshots of fixed conformations. Consequently, dynamics must be inferred, missing vital information describing how the biomolecules truly function in native conditions, such as their fluctuations, rates, intermediate states and statistical distributions. Nuclear magnetic resonance (NMR) spectroscopy provides both structural and dynamic information on biomolecules but is currently suited to smaller soluble proteins and picosecond to nanosecond timescale dynamics of specific sites[6].

A number of different light microscopy techniques can observe dynamics of single molecules. However, despite significant improvements in the localization resolution with methods such as stimulated emission depletion microscopy (STED)[7] and stochastic optical reconstruction microscopy (STORM)[8,9], the imaging resolution is not able to go below ~20 nm[10]. Such resolution does not allow protein-protein interactions to be directly observed, nor does it enable structural features or dynamics to be assessed. A method that is sensitive to less than 10 nm with a time resolution of typically ~10 ms is fluorescence resonance energy transfer (FRET). The spatial resolution of FRET is dependent on the Förster radius of the pair of fluorescent molecules between which energy is transferred[11]. FRET is sensitive to distance changes as small as 0.3 nm in the 3–10 nm inter-dye distance range[12]. However, reducing the Förster radius also reduces the technique's sensitivity range, limiting it to site specific interactions over specific spatial windows. A technique that can access nanosecond timescales is fluorescence correlation spectroscopy (FCS)[13]. By measuring intensity fluctuations as fluorescent molecules diffuse in and out of a detection volume, FCS can determine concentrations, mobility and interactions of labeled molecules. Spatially however, FCS is limited by the diffraction limit to hundreds of nm resolutions and can suffer from poor autocorrelation signal-to-noise ratio at high molecular densities. The spatial resolution can be improved to as low as 30 nm using a combination of methods such as FCS-STED, however, this is often at the expense of lower temporal resolution[14]. Similarly, the temporal resolution of FRET has been improved to sub-millisecond time scales using diffusion-based FRET to detect one molecule at a time as it freely diffuses in solution. However, in this condition the length that a single molecule can be measured is greatly reduced to ~10 ms[15].

Whilst many of these techniques can provide valuable insight into biomolecular processes, few can simultaneously provide structural and dynamical information of single molecules on microsecond timescales, and none can provide microsecond time resolution over seconds or minutes of observation. Additionally, these techniques require labeling of molecules that can modify the very dynamics of interest. High-speed AFM (HS-AFM) offers a label-free technique that has submolecular imaging resolution with high spatiotemporal resolution, ~1 nm lateral, ~0.1 nm vertical and ~100 ms temporal resolution. Although HS-AFM proves to be a valuable tool in understanding the function and behavior of many proteins at the single molecule level[16–18], there are many molecular processes that are too fast to be resolved in imaging mode[19,20]. Progress in developing faster HS-AFM is ongoing but it may be unlikely to reach sub-millisecond imaging resolution in the near future.

Here, inspired by fluorescence spectroscopy, we develop and apply HS-AFM height spectroscopy (HS-AFM-HS), a technique whereby we hold the AFM tip at a fixed $x$–$y$ position and monitor the height fluctuations under the tip in $z$-direction with Angstrom spatial and 10μs temporal resolution. We demonstrate how this technique can be used to simultaneously measure surface concentrations, diffusion rates and oligomer sizes of highly mobile annexin-V molecules during membrane-binding and self-assembly at model membranes and derive its kinetic and energetic terms. Additionally, HS-AFM-HS at specific positions in the annexin lattice where the freedom of movement is restricted to rotation allowed determination of the interaction free energies of protein-protein contacts. The applicability of our technique is wide and is discussed in the end of the manuscript.

## Results

**Reduced dimensionality leads to ms and μs HS-AFM.** Annexin-V has been shown, among other functions[21], to play an important role in membrane repair of eukaryotic cells[22]. The influx of $Ca^{2+}$ from the outside of the cell that occurs upon membrane lesion leads to the rapid tripartide Annexin-V-$Ca^{2+}$-membrane-binding and self-assembly of Annexin-V into 2D-crystals, surrounding the membrane defect to prevent further pore expansion[22]. Interactions between annexins, negatively charged phospholipids and $Ca^{2+}$ have been the subject of many studies[23–25]. In solution Annexin-V alone has been shown to have a low $Ca^{2+}$ affinity (~330 μM), whereas in the presence of a negatively charged phospholipids the $Ca^{2+}$ affinity increases greatly with two distinct affinities (~2.4 μM and 170 μM)[26,27]. Whilst binding and final assemblies have been well characterized[27,28], no techniques are able to capture the full process and bridge between binding to the membrane, oligomerization and 2D self-assembly into a functional lattice structurally and quantitatively.

HS-AFM imaging of supported lipid bilayers (SLBs) containing 20% phosphatidylserine (Fig. 1a) shows how annexin binding to the surface of the membrane (upper leaflet) and subsequent self-assembly occurs over the second timescale (Fig. 1b). Assembly of Annexin-V from solution was initiated by illuminating the sample with UV to release $Ca^{2+}$ from a photo-cleavable EGTA-$Ca^{2+}$-complex (Fig. 1b, c). Annexin was observed binding to the membrane within a few seconds, reaching full membrane coverage after 32 s and 2D-crystal ($p6$-symmetry)[27] formation within 40 s. However, 2D-scanning is not able to resolve the

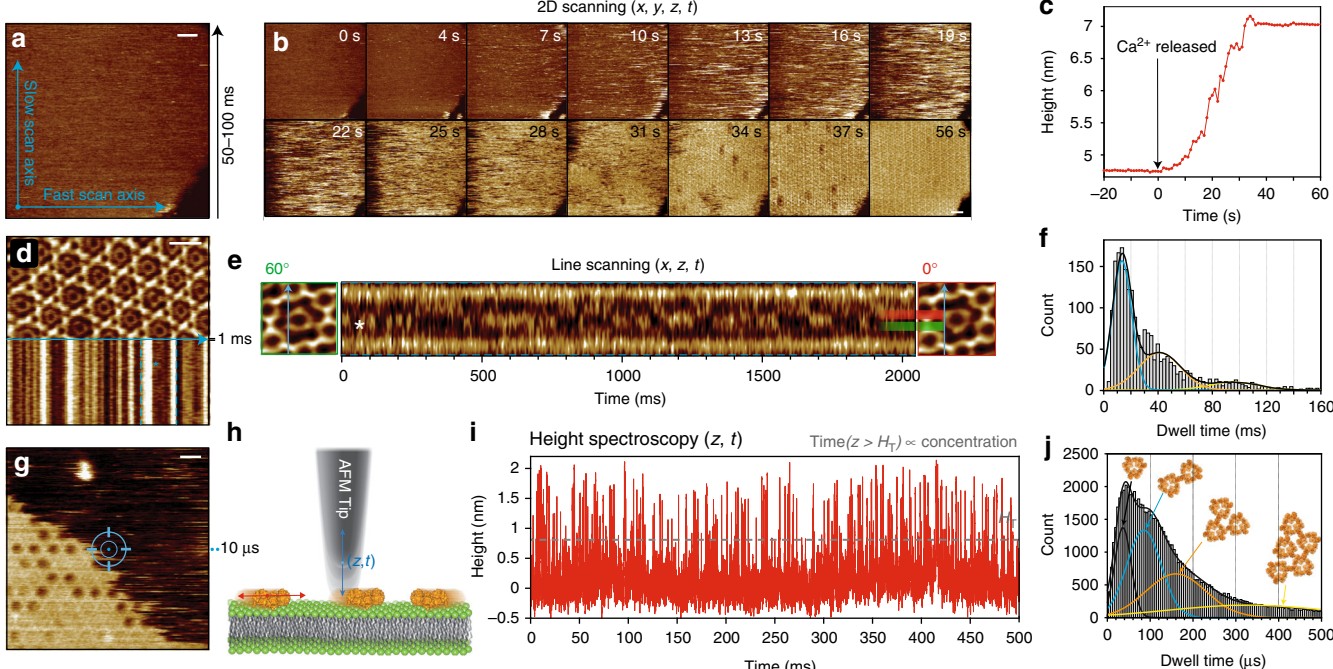

**Fig. 1** Increasing the temporal resolution of HS-AFM by reducing the dimensionality of data acquisition. **a** HS-AFM image of a DOPC/DOPS (8:2) membrane in the presence of annexin-V and NP-EGTA-caged $Ca^{2+}$. Blue arrows illustrate the slow- (vertical) and the fast-scan axis (horizontal). Images can be captured at up to 10–20 frames $s^{-1}$. **b** HS-AFM movie frames of A5 membrane-binding, self-assembly and formation of $p6$ 2D-crystals upon UV-illumination induced $Ca^{2+}$-release. **c** Average height/time trace of the membrane area in **b**. **d** Averaged HS-AFM image of an A5 $p6$-lattice overlaid with the subsequent line scanning kymograph, obtained by scanning repeatedly the central $x$-direction line as illustrated by the blue arrow with a maximum rate of 1000–2000 lines $s^{-1}$. **e** Line scanning kymograph across one protomer of the non-$p6$ trimer, marked by * in **d** and **e** at a rate of 417 lines $s^{-1}$ (2.4 ms per line). **f** Histogram of state dwell-times of the molecule in **e**. **g** HS-AFM image of an A5 $p6$-lattice partially covering a DOPC/DOPS (8:2) SLB surface during self-assembly. HS-AFM height spectroscopy (HS-AFM-HS) is performed following halting the $x$- and $y$-piezos to capture height information at a fixed position at the center of the image (illustrated by the target). **h** Schematic showing the principle of HS-AFM-HS. The AFM tip is oscillated in $z$ at a fixed $x,y$-position, detecting single molecule dynamics such as diffusion under the tip. **i** Height/time trace obtained by HS-AFM-HS with the tip positioned at the center of image (**g**). The height/time trace allows determination of the local A5 concentration analyzing the time fraction of the occurrence of height peaks. **j** Dwell-time analysis of each height peak of diffusing A5 from 60 s height/time data and subsequent fitting of the distribution to multiple Gaussians (possible molecular aggregates corresponding to the fits with distinct dwell-times ($\tau_D$) are shown above the graph). All scale bars: 20 nm

highly mobile membrane-bound annexin trimers (A5) during the early stages of the assembly process, as they diffuse too quickly to be resolved when images are acquired at frame rates of $1\,s^{-1}$ to $10\,s^{-1}$; instead, only streaks in the fast scanning direction ($x$) are observed (herein A5 is used to refer to the membrane-bound trimeric form of annexin-V). The average height and abundance of these streaks across the membrane patch can be used to approximate the surface coverage over time (Fig. 1c); however, such measurements are prone to error due to tip parachuting (where the tip loses contact with the sample and takes some time to return to the surface) and tip induced movement of the proteins. The maximum HS-AFM imaging rate for a typical 100×100 pixel 2D-scan is 50–100 ms. Each image is scanned left to right and right to left, thus at 50 ms per image, 400,000 pixels at which deflection and height are detected are acquired per second (one every 2.5 µs).

To obtain 100 times improved temporal resolution the slow-scan axis ($y$-direction) is halted (Fig. 1d) and fast-scan axis lines ($x$-direction) are acquired every 0.5–1 ms. This produces single line height data ($x$, $z$, $t$), termed line scanning (HS-AFM-LS), in which all of the traces in the $x$-direction are over the same $y$-position. One A5, at the $p6$-axis of the annexin-V 2D-crystal, is free to rotate and has two preferred orientations (Fig. 1e, inset)[27]. Performing HS-AFM-LS across the annexin lattice (Fig. 1d, bottom) we can visualize A5 rotation (Fig. 1e). The rotation is most evident for those trimers where the scan line crosses

precisely one protomer of the trimer (Fig. 1d, e; highlighted with *). Capturing lines at 2.4 ms per line over several seconds shows negligible drift in $x$–$y$, as observed by the two stable lines (Fig. 1e, at $x = 4$ nm and $x = 16$ nm) originating from the stable neighboring annexins in the $p6$-lattice. By contrast, the central region shows rapid flickering between two states above and below $x = 10$ nm (Fig. 1e). Comparison to a model line-scan across a rotating trimer shows identical switching behavior caused by clockwise and counter-clockwise rotations (Supplementary Fig. 1). Analysis of the periods of time spent in each state before rotation (Fig. 1f) shows a wide distribution best fit by three Gaussians (as determined by reduced chi-squared values) peaking at 13, 41 and 96 ms, suggesting possibly three different modes of interaction with the surrounding lattice, that we tentatively assign to the three possible interaction sites of the rotating trimer with its environment.

Whilst HS-AFM-LS provides single-digit millisecond temporal resolution, it is still not fast enough to capture microsecond events. Therefore, to gain a further 100-fold time-resolution we also halt the $x$-piezo to capture Angstrom accuracy height data ($z$, $t$) with ~10 µs temporal resolution (Fig. 1i, Supplementary Fig. 2). This method, termed high-speed AFM height spectroscopy (HS-AFM-HS, Fig. 1h), has the temporal resolution to measure the mobility of membrane-bound molecules as they diffuse under the tip. A typical height vs. time trace obtained at the surface of an SLB with A5 diffusing on the surface (Fig. 1i) gives a signal time

series of many sharp peaks of up to ~2 nm height, corresponding well to the height of membrane-bound A5. A distribution of heights between $H_T$ and ~2 nm is observed due the z-feedback not being able to fully respond to the shorter dwell-times (Supplementary Fig. 2c). For future applications of even faster events, the amplitude damping of the cantilever oscillation can be monitored, which should report about events beyond the feedback bandwidth. This data was captured at the tapping frequency of the cantilever 625kHz (1.6 µs), with feedback settings set to maximize the z-piezo response time (Supplementary Fig. 2). Measuring the time duration of each peak above $H_T$, gives a distribution of dwell-times corresponding to the range of times molecules spend under the tip (Fig. 1j), with the fastest events being only ~10 µs long.

**A5 diffusion and oligomerization measured by HS-AFM-HS.** For proteins undergoing 2D Brownian diffusion, the dwell-time $\tau_D$, of the molecule in a detection area is dependent on the protein's diffusion coefficient ($D$), and the width of the detection area $w$, by Eq. (1)[29].

$$\tau_D = w^2/4D \qquad (1)$$

For techniques such as FCS, the detection area is defined by a fluorescence spot size much larger than the molecules of interest, leading to the protein dimensions having a negligible contribution to $\tau_D$. For HS-AFM-HS however, the inverse is true; the detection area, which is essentially the AFM tip radius (~1 nm; we know this because substructures on the proteins can be resolved, see Fig. 1d), is typically much smaller than the size of the single diffusing proteins (~10 nm diameter) and thus dwell-times are mainly dependent on the molecule size. For molecules undergoing self-assembly the increase in the 2D-area is thus linear with each additional molecule $n$ associated to the aggregate, and therefore $\tau_D$ increases proportionally with $n$. Performing height spectroscopy on A5 molecules undergoing self-assembly into higher-order oligomers is therefore expected to produce the multi-peaked distribution of dwell-times we observe in Fig. 1j, which not only depends on oligomer size but also its size-dependent diffusion rate (see Supplementary Table 1 for full details of oligomer dimensions). The $\tau_D$ distribution is well approximated by Gaussian fits however the exact underlaying distribution is expected to be Lévy in nature with heavy tailed probability distributions. The diffusion coefficient $D$ can then be determined from the expected protein dimensions, tip radius and $\tau_D$ (Supplementary Table 1).

In addition to the oligomer size and diffusion characteristics we can also measure the surface concentration of protein, based on the probability that a molecule is present under the tip at any given time. This probability can be determined by the fraction of time the height $z$, is above a threshold value $H_T$ ($t_{z>HT} / t_{total}$), and converted into a surface density $c$ (molecules µm$^{-2}$) based on the molecule size $d_p$, via the following relation:

$$c = \frac{t_{z>H_T}}{t_{total}} \cdot \frac{1}{d_p^2} \qquad (2)$$

The threshold height $H_T$ was not an arbitrary value but chosen based on the background noise level of the height trace, significantly far away from the noise distribution at $5\sigma$ so that the probability of mistaking diffusion events is 0.00006% (typically this corresponds to $H_T = 0.8 \pm 0.1$ nm (s.d.)).

To assess the kinetics of the 2D-assembly process of A5 at membranes, we varied the bulk concentration of calcium with a fixed annexin-V solution concentration. HS-AFM-HS on hydrated SLBs in the absence of Ca$^{2+}$ (Fig. 2a) gives a random height noise trace that fluctuates with typical RMS amplitude of 0.17 nm, sampled at cantilever resonance frequency of 625kHz. The introduction of 50 µM CaCl$_2$ to the bulk phase (Fig. 2b) resulted in a small number of infrequent (~6 s$^{-1}$) sharp peaks above noise corresponding to single molecule diffusion events under the tip. Assessment of the time fraction gave an A5 surface concentration of 1.0 molecules µm$^{-2}$ ± 0.6 (s.d.) whilst analysis of individual dwell-times gave a distribution with a dominant Gaussian peak at ~33 ± 26 µs (s.d.), with small and less significant peaks at ~80 µs and ~125 µs. This $\tau_D$ distribution implies a dominant species diffusing on the membrane with a diffusion coefficient $D$ of $0.8 \pm 0.6$ µm$^2$ s$^{-1}$, assuming the trimeric form (A5) of membrane-bound annexin. This assumption can be made based on previous studies which suggest that annexin molecules exist in monomeric form only in solution, forming stable trimers almost instantaneously in the presence of Ca$^{2+}$ and anionic lipids as they bind to the membrane[30]. Additionally, the expected dwell-time under the tip for a single protomer diffusing at 1 µm$^2$ s$^{-1}$ would be 2.7 µs, outside the z-feedback response time and therefore would not be detected. Inversely, a dwell-time of 33 µs for a single annexin-V protomer would imply an unrealistically slow diffusion coefficient of 0.08 µm$^2$ s$^{-1}$ (for the full molecular diffusion/size range currently accessible by HS-AFM-HS see Supplementary Fig. 3). A5 diffusion has previously been shown to be of the order 1 µm$^2$ s$^{-1}$ using FRAP[31], in good agreement with the $0.8 \pm 0.6$ µm$^2$ s$^{-1}$ found here.

As the Ca$^{2+}$-concentration was increased from 50 µM to 100 µM, 150 µM and 200 µM (Fig. 2c–e) we observed increases in both the frequency and dwell-times of events; equating to a three orders of magnitude increase in the surface density of A5 from $1.0 \pm 0.6$ to $285 \pm 150$ A5 µm$^{-2}$. The increase in surface density can also be seen qualitatively by the occurrence of streaks in HS-AFM images because the molecules diffuse too fast for the HS-AFM to capture whilst 2D-scanning. Analysis of the height spectroscopy events shows the emergence of additional Gaussian peaks at $80 \pm 25$ µs and $130 \pm 34$ µs. Taking the additional peaks to be dimers (A5$_2$) and trimers (A5$_3$) of A5, we can determine A5$_2$ and A5$_3$ diffusion coefficients of $0.63 \pm 0.21$ µm$^2$ s$^{-1}$ and $0.58 \pm 0.16$ µm$^2$ s$^{-1}$, respectively, which—as expected for larger molecules—is less than the A5 diffusion coefficient of 0.8 µm$^2$ s$^{-1}$. We observe small shifts of all peaks to longer dwell-times as a function of the Ca$^{2+}$-concentration, eg the primary A5 peak shifts from ~33 µs to ~37 µs and the secondary A5$_2$ peak from ~78 µs to ~85 µs, which we interpret as the result of crowding when the 2D density of A5 increases on the membrane leading to a slow-down of the diffusion rates.

Increasing the Ca$^{2+}$-concentration to 250 µM (Fig. 2f) resulted in the onset of 2D-crystallization. This can be observed both in imaging mode, as a $p6$-lattice 2D-crystal partially covering the membrane, and in height spectroscopy mode, by the much longer-lived events, which last several tens of milliseconds, as the crystal assembly and disassembly is detected under the tip. Under these conditions, in addition to the three peaks at ~33 µs, ~80 µs and ~130 µs detected at lower Ca$^{2+}$-concentrations (Fig. 3b–e), significantly larger peaks at longer dwell-times at $200 \pm 60$ µs and $315 \pm 100$ µs are detected (Fig. 3f, right). Because the dwell-time distributions are short and almost mono-disperse at very low surface concentrations (Fig. 2b, c, right), and are more and more convoluted with increasing bulk Ca$^{2+}$-concentration and increasing A5 surface concentration (Fig. 2d–f, right), we assign the underlying peaks to A5, dimers of trimers A5$_2$, trimer of trimers A5$_3$, and so on, A5$_4$, A5$_5$, and higher-order oligomers (A5$_o$), and thus use these molecular dimensions to determine the oligomer size-dependent diffusion coefficients (Supplementary Fig. 4). Diffusion constants, derived from the dwell-time peaks, show a

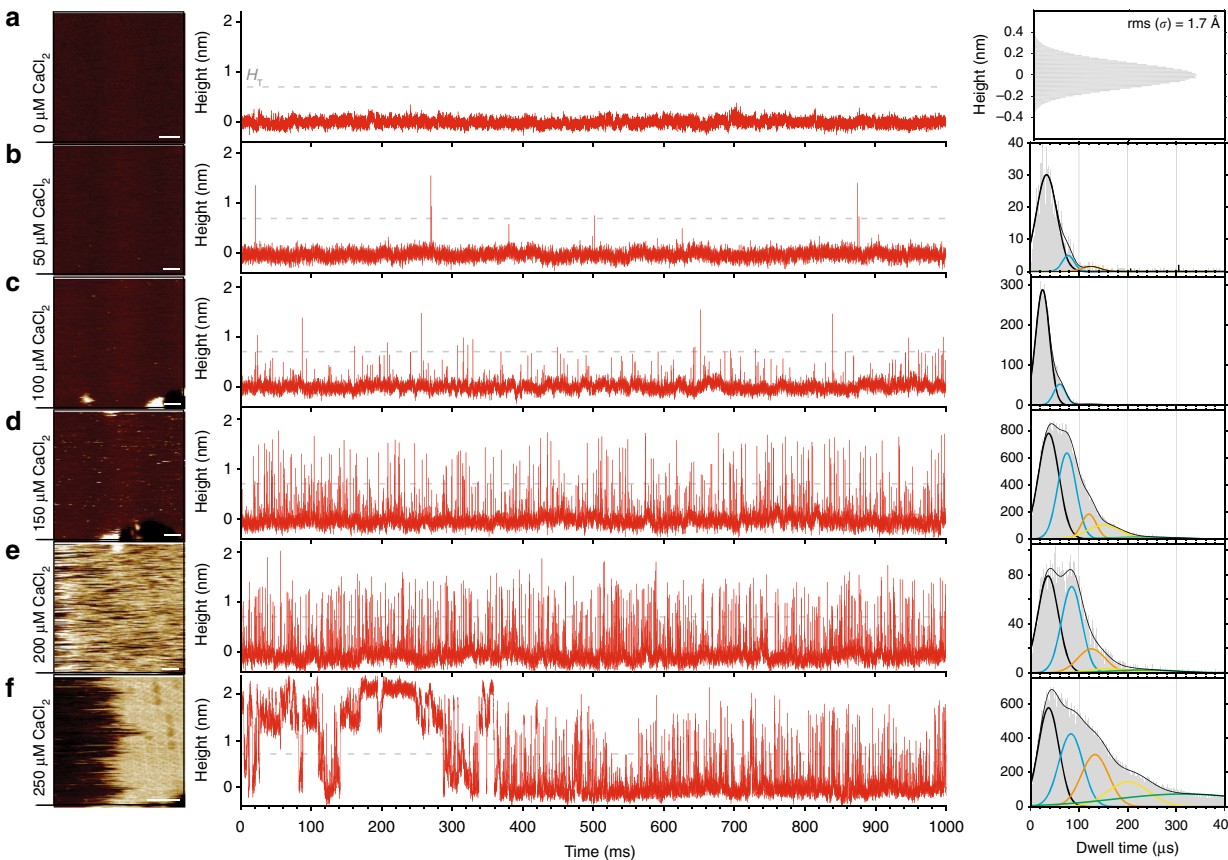

**Fig. 2** HS-AFM-HS of diffusion and self-assembly at model membranes. **a** HS-AFM image frame (left) and height spectroscopy height/time trace (middle) of the membrane (8:2 DOPC/DOPS) surface without $Ca^{2+}$ in solution. The histogram (right) shows the full distribution of height values detected indicating a noise level of 0.17 nm at 625 kHz sampling rate. **b**–**f** HS-AFM image frames (left) and subsequent height spectroscopy height/time traces (middle) at the membrane surface with 50 μM (**b**), 100 μM (**c**), 150 μM (**d**), 200 μM (**e**) and 250 μM (**f**) $Ca^{2+}$ in solution. Right: corresponding dwell time ($\tau_D$) distributions with multi-component Gaussian fits. All measurements were performed in the presence of 127 nM annexin-V in solution. Images: Full color scale: 4 nm, scale bars: 50 nm. 1000 ms height/time traces are example sections from longer, typically 60 s, traces

decrease with increasing oligomer size, consistent with the Saffman–Delbrück approximation[32].

$$D \sim \ln\left(1/r_p\right) \qquad (3)$$

Since the height/time traces have only Angstrom-range noise (Fig. 3a), the occurrence of molecular diffusion events with varying dwell-times (Fig. 3b–d) are unambiguously detected and allow us to determine what oligomeric species are present and at what abundance dependent on the environmental conditions or the overall 2D-concentration on the membrane. The changes in overall surface concentration and oligomer distribution with varying $Ca^{2+}$-concentration (Fig. 3e) indicate that in the presence of up to 100 μM $Ca^{2+}$, annexin-V molecules are predominantly in the trimeric A5 form. At higher $Ca^{2+}$ (150–200 μM), significant fractions of A5 encounter other A5 and convert into dimers of trimers $A5_2$. At calcium concentrations that permit 2D-crystallization (250 μM), we observe reductions in the fractions of both the A5 and $A5_2$ populations with significant increases in the fraction of trimers of trimers $A5_3$ and larger oligomeric structures $A5_o$.

The total fraction of time molecules spend under the tip during a given period allows the determination of surface concentration. Here, the average surface concentration of A5 grew exponentially with bulk $Ca^{2+}$-concentration (Fig. 3e). The determined surface concentration combined with knowledge of the bulk

concentration allows calculation of a partition coefficient, $P = [c_{\text{solution}}]/[c_{\text{surface}}]$, and hence the free energy associated with annexin-V binding to the membrane, following $\Delta G_0 = k_B T \cdot \ln(P)$. These calculations give free energies that decrease with increasing calcium concentration (indicating stronger binding), with values of $-1.7 k_B T$, $-3.1 k_B T$ and $-6.1 k_B T$ at 50 μM, 100 μM and 150 μM $CaCl_2$ respectively, reaching a minimum of $-10.9 k_B T$ at 250 μM $CaCl_2$ in agreement with previous studies[26].

Next, we investigated the binding of A5 depending on variations of the bulk annexin-V concentration in the presence of saturating 2 mM $Ca^{2+}$-conditions (Supplementary Fig. 5). Analogous to the A5 surface binding behavior at varying $Ca^{2+}$-concentrations, the average surface concentration increased exponentially with bulk annexin-V concentration, and higher oligomeric states accumulate (Fig. 3f). Binding free energies, as determined by partition coefficients in saturating $Ca^{2+}$, decrease with increasing bulk annexin-V concentrations from $-4 k_B T$ and $-6 k_B T$ at 23–35 nM and 58–81 nM annexin-V respectively, reducing to $-7.7 k_B T$ at 103 nM before reaching a minimum of $-10.9 k_B T$ at 127 nM annexin-V. When the data from all bulk annexin-V concentrations is combined (Fig. 3g) we observe how the average dwell-times change from dilute surface concentrations, in which the average dwell-time increases gradually with surface concentration, to higher surface concentrations, where the dwell-times increase more rapidly with surface concentration. This transition occurs at approximately 500 A5-molecules μm$^{-2}$

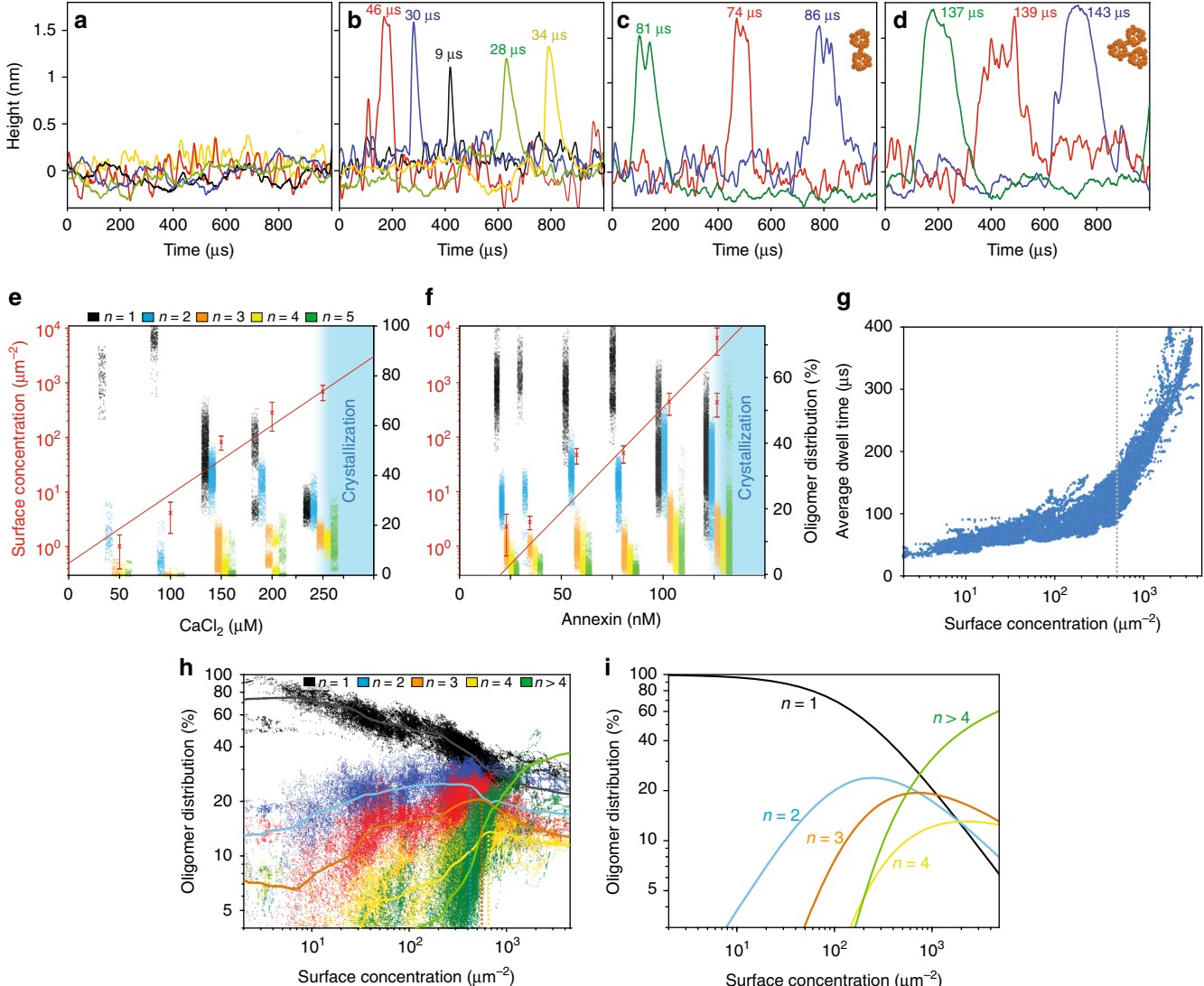

**Fig. 3** Determination of A5 oligomeric states and dynamics on the membrane. **a–d** Overlaid 1 ms height/time traces showing background noise (**a**) and diffusion event peaks around 33 μs (**b**), 80 μs (**c**) and 130 μs (**d**) dwell-times. (**e**) Averaged surface concentrations (mean±s.d.) and oligomer distributions obtained by dwell-time peak fitting for varying Ca²⁺-concentrations at constant 127 nM annexin-V bulk concentration, and (**f**) at varying annexin-V bulk concentrations at constant 2 mM Ca²⁺ concentration. Shaded areas in **e** and **f**: concentrations at which self-assembly into 2D-crystals is observed. **g** Graph displaying the average A5 diffusion dwell-times under the tip as a function of surface concentration. Average dwell-times are calculated from dwell-times observed within 1-second time-windows. At ~ 500 molecules μm⁻² surface density (dashed line), the average dwell-times changes regime indicative of the formation of higher-order oligomers. **h** Experimental and **i** theoretical oligomer distributions as a function of surface concentration. Oligomer distributions in **h** and surface concentrations (x-axes in **h** and **g**) are calculated over 1-second time-windows during the height spectroscopy analysis. The oligomer distributions are determined based on the range covered by the dwell-times peaks for each oligomeric state (see Fig. 2, right). Theoretical oligomer distributions were calculated using experimentally fitted equilibrium binding constants (Table 1) between A5 and higher-order assemblies, A5₂, A5₃, etc

(~3% surface coverage) (Fig. 3g, dashed line and kink in the data point distribution). We interpret this in the following way: at low surface concentrations, molecules diffuse freely predominantly in their A5 state. The minor dwell time increase may be related to a slight slow down of diffusion due to the onset of crowding. However, when a critical surface concentration is reached (~3%) the encounter probability increases significantly and protein-protein interactions become significant, and higher oligomers are formed. For experiments in varying Ca²⁺ concentrations, fitting to the A5 surface concentration vs. Ca²⁺ bulk concentration (Fig. 3e) show this critical A5 surface coverage is reached at 240 ± 10 μM Ca²⁺, in agreement with previous work on model membranes and cells[26,33]. Assignment of the dwell-time distribution peaks (Fig. 2, right column) to oligomer sizes allows the

populations of each oligomer species to be investigated and plotted as a function of surface concentration (Fig. 3h). As the A5 surface concentration increases, the fraction of A5 molecules in a single trimer form decreases, whilst the fractions of assemblies composed of 2 (A5₂), 3 (A5₃), 4 (A5₄) and higher oligomeric states (A5ₒ) each increase at successively higher surface coverages. Near the critical surface concentration (500 molecules μm⁻², ~3% surface coverage), the populations of the A5₂, A5₃ and A5₄ reach maximal fractions at successive surface concentrations before decreasing and giving rise to higher-order oligomers, which become the dominant population as the 2D-lattice begins to form.

Oligomerization of A5 at the membrane is a 2D reaction in which the concentration of each oligomer species, [AB] is a

function of the surface density of its component parts, [A] and [B]. Thus, under equilibrium conditions, the kinetics can be described by 2D-dissociation constants, $K_d$.

$$[AB] = [A][B]/K_d \qquad (4)$$

For instance, the dimer dissociation constant, $K_{d2} = [A5] \cdot [A5]/[A5_2]$, is defined in terms of the equilibrium surface densities of A5, and $A5_2$. Whilst the trimer dissociation constant, $K_{d3} = [A5] \cdot [A5_2]/[A5_3]$, is dependent on the A5, $A5_2$, and $A5_3$ concentrations. Fitting oligomer concentration data (Supplementary Fig. 6) to Eq. (5) allows experimental determination of $K_d$.

$$\text{Fraction of } [B] \text{ in complex} = \frac{[AB]}{[AB] + [B]} = \frac{[A]}{[A] + K_d} \qquad (5)$$

The $K_d$ values obtained from these fits are organized into an $AxB$ matrix for the resulting oligomers of size [AB] (Table 1). The dissociation constants $K_{dn}$ obtained for the formation of $A5_2$, $A5_3$, and $A5_4$ are comparable and average to $250 \pm 70 \, \mu m^{-2}$ (+ 95% CI) suggesting similar interaction strengths between different oligomers.

Computation of the surface concentration dependent populations of each oligomer species using the experimentally determined $K_d$ values (Supplementary Fig. 6) shows how the fractions of higher-order oligomers change in a stepwise manner (Fig. 3i). As the overall surface concentration increases, the fraction of monomers decreases, followed by peaks in population for $n = 2$ (at concentrations close to $K_{d2} = 220 \, \mu m^{-2}$), $n = 3$ and $n = 4$ as the population of each new higher-order structure $(n+1)$ is able to assemble depending on the abundance of the previous one. These characteristics are in close agreement with the experimental populations (Fig. 3h).

From ratios between different oligomer species it is possible to estimate a free energy difference between oligomer states, $\ln(c_n / c_m) = \Delta G/k_B T$ (Supplementary Fig. 7), where $c_n$ and $c_m$ are the surface concentrations of oligomers constituted of n and m A5s, respectively.

**Rapid A5 membrane-binding precedes oligomer assembly**. By using the photosensitive caging compound NP-EGTA to cage calcium we can simulate the burst of $Ca^{2+}$ that would occur in a cell when the plasma membrane was injured, to follow the annexin binding and oligomerization over time[27] to an initially bare membrane (Fig. 4a). Upon UV-illumination, we observed under the here used experimental conditions (1 mM caged-$Ca^{2+}$ and 200 nM annexin-V in solution) single A5 diffusion events after 2.8 s (Fig. 4b, d–f). In the beginning, the frequency of events increased linearly with time (Fig. 4g, red line) before more dramatically increasing and fluctuating as a function of time, reaching a maximum of $2600 \, s^{-1}$ after 73 s illumination, when 2D-crystallization sets in. Additionally, the average residence time under the tip of each event, initially relatively constant around 40 µs, in agreement with single A5 diffusion (Figs. 2 and 3b), increased and fluctuated similarly to the event count rate as the membrane gets crowded, in agreement with the formation of higher-order oligomers. Surface concentration measurements show a gradual increase over time (Fig. 4h, red line) before sharply increasing as a critical concentration of 400–600 molecules $\mu m^{-2}$ is reached, in agreement with the pooled data equilibrium experiments (see Fig. 3g). Analysis of the oligomer distribution over time indicates that this sharp increase coincides with a sudden onset in the fraction of oligomers composed of 5 or more A5. After 100 s UV-illumination, HS-AFM imaging showed

**Table 1 $K_d$ values arranged in a matrix where each molecular species arises from the sum of the components parts**

| $K_d$ ($\mu m^{-2}$) | Monomer | Dimer | Trimer | Tetramer |
|---|---|---|---|---|
| Monomer | 220 ± 40 | 170 ± 20 | 650 ± 60 | 200 ± 10 |
| Dimer | 152 ± 31 | 444 ± 60 | 210 ± 20 | 128 ± 8 |
| Trimer | 260 ± 50 | 210 ± 20 | 180 ± 20 | 109 ± 10 |
| Tetramer | 170 ± 30 | 106 ± 21 | 91 ± 18 | 52 ± 10 |

$K_d$ values were obtained by fits to Eq. (5) (Supplementary Fig. 6) (±95% CI). $K_d$ values for forming complexes larger than 5 units such as the tetramer-tetramer interaction, have reduced reliability as all molecular aggregates above $n = 4$ are taken together in the fitting

the resulting complete p6-lattice of A5 with no apparent perturbation caused by the HS-AFM-HS measurement (Fig. 4c).

**Single A5 rotational dynamics revealed by HS-AFM-HS & -LS.** Once the A5 p6-lattice is assembled, it contains two-thirds p6-trimers that constitute the honeycomb lattice and one-third non-p6 trimers that are not strictly part of the lattice being trimers sitting on the 6-fold symmetry axis (Fig. 5a, see also Fig. 1d). The non-p6 trimers only weakly interact with the p6-lattice at two preferred orientations at 0° and 60° (Fig. 5b)[27]. The interaction is weak enough such that it allows rotational freedom intermittently resolvable by HS-AFM imaging (Fig. 5a, Supplementary Movie 1). Measuring this rotational freedom provides a means to determine non-p6-trimer interactions with neighboring molecules and directly compare dynamics observed by line scanning with HS-AFM-HS. Positioning the tip on one of the protomers of a non-p6 trimer and performing HS-AFM-HS gives a height trace that fluctuates over time between two distinguishable states with heights of $2.00 \pm 0.10 \, nm$ (mean±s.d.) and $1.72 \pm 0.07 \, nm$ (Fig. 5c). Performing HS-AFM-HS on immobile trimers in the hexagonal p6-lattice produced a height trace with only one state (±0.12 nm). It should be noted the scanner stage can drift by some nanometers in x- and y-dimensions, especially during the seconds after execution of a position or scan range change due to piezo-relaxation. To assess the mechanical drift, we can capture HS-AFM images of the A5-lattice for several minutes and then use image correlation alignment software to find the x–y translations required to align the image set. Under normal imaging conditions the total drift distance was found to vary from as low as $0.02 \, nm \, s^{-1}$ with a well-equilibrated system (as shown in Supplementary Fig. 8, Supplementary Movie 2) up to $0.1 \, nm \, s^{-1}$. This drift is relatively slow ($10–50 \, s \, nm^{-1}$) in comparison to the tip radius, the area of interest and the biological dynamics to be analyzed ($\gg 10 \, s^{-1}$) and suggests that HS-AFM-HS can be positionally accurate for tens of seconds. Such stability provides an additional advantage of HS-AFM-HS and HS-AFM-LS over fluorescence techniques where bleaching often limits the total time a molecule can be observed with high temporal resolution.

Performing line scanning across the non-p6 trimer (as depicted by the dashed line in Fig. 5a, b) produces kymograph images displaying time (x-axis), position (y-axis) and height (color scale) (Fig. 5d, top). The line scan kymograph detects the non-p6 trimer primarily at one of two positions ~ 3 nm apart in y as shown by the labels marked 0° and 60°. Over time flickering between the two states is observed. Plots show how the heights in these positions fluctuate around ~1.8 nm and ~2.0 nm over time for both the 0° and 60° positions (Fig. 5d, green and red traces). These height changes are closely comparable to those obtained by HS-AFM-HS (Fig. 5c). However, because the line scanning measurement captures data at two different regions of interest that behave in an anti-correlated manner with each other, taking the difference between the two height traces can be used to

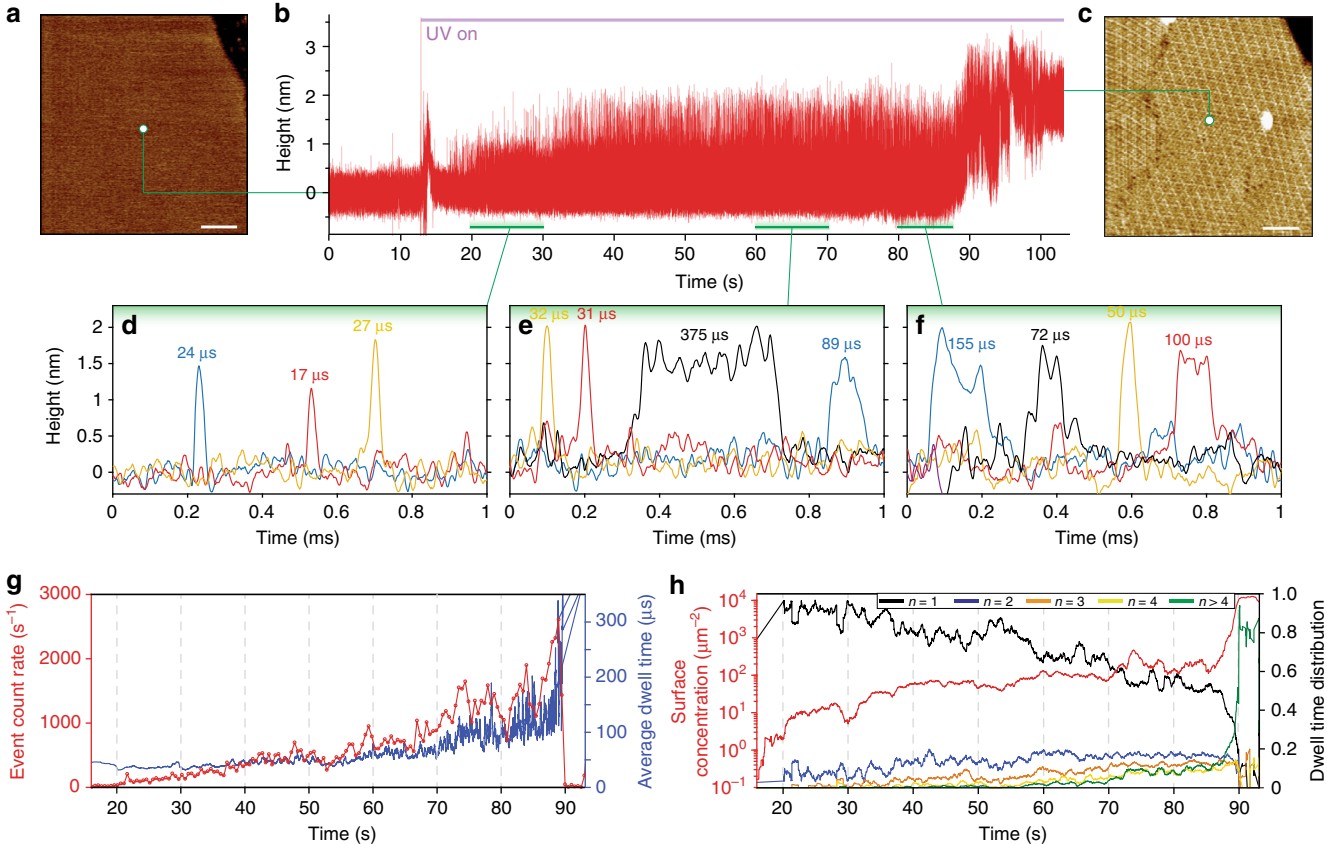

**Fig. 4** Time-lapse HS-AFM-HS of annexin-V membrane-binding and self-assembly. **a** HS-AFM image taken before A5 membrane-binding and self-assembly (bare lipid bilayer). **b** HS-AFM-HS height/time trace during illumination with UV-light to release $Ca^{2+}$. **c** HS-AFM image taken directly after **b** (*p6*-lattice 2D-crystals). **d–f** Show higher temporal resolution zoom-ins of the HS-AFM-HS trace (**b**) showing example diffusion events from the 20–30 s, 60–70 s and 80–87 s time regions, respectively, with different line colors representing different events. **g** Number of single diffusion events (red) and averaged dwell-times (blue) over time. **h** Overall surface concentration (red) and oligomer species (black: A5, blue: $A5_2$, orange: $A5_3$, yellow: $A5_4$, green: $A5_O$) distribution changes over time. Traces in **d**, **e** are averaged over 1 s time-windows (time scale panels **c**, **d** and **e** are matched). Scale bars: 50 nm

amplify the signal whilst reducing any correlated errors that occur in both traces. This produces an enhanced signal-to-noise ratio (Fig. 5d, black trace) with two distinct states (Fig. 5g). The movements between the 0° and 60° states, detected by HS-AFM-HS and line scanning result in average dwell-times of $26 \pm 3$ ms (s.e.m., $n = 401$) and $35.0 \pm 1$ ms (s.e.m., $n = 2053$), respectively (Fig. 5f). This difference, measured by the two methods, is likely due to the greater energy input via the tip in HS-AFM-HS combined with the higher temporal resolution of HS-AFM-HS which was able to detect dwell-times as short as 240 µs, beyond the time resolution of line scanning.

Although HS-AFM-HS offers higher temporal resolution, the ability of line scanning to also measure position allows direct visualization of the rotational velocity of the trimer (Fig. 5e). Line scanning kymographs reveal linear transitions in time from the 0° to the 60° or *vice versa*. As visible in raw data (Fig. 5e, top) and corroborated by model line scanning rotations (not fitted to the data) (Fig. 5e, bottom), the characteristics of these transitions depend on the initial state and the direction of rotation (Supplementary Fig. 1). Rotations were observed to occur in both clockwise and counter-clockwise directions with an average time of $18 \pm 6$ ms, corresponding to a rotational velocity of $3300°\mathrm{s}^{-1}$ (550rpm) (Fig. 5e, right). As expected by the symmetry of the system, the non-*p6* trimer showed no preference for either the 0° or 60° state and as such, there is no free energy difference between states. We can however estimate a free energy barrier that is overcome between the 0° and 60°

states of $\sim 0.7k_BT$ ($\pm 0.4k_BT$ (s.d.)), using the average time spent in each state and the average time the trimer is rotating, by the following relation:

$$\Delta G = -\ln(\tau_{\mathrm{rotation}}/\tau_{\mathrm{state}})k_BT \qquad (6)$$

## Discussion

In this work, we have developed and applied two HS-AFM techniques, HS-AFM line scanning and HS-AFM height spectroscopy (HS-AFM-HS), which allow Angstrom-precision dynamic measurements of single molecules at millisecond- and $\sim 10$ µs-timescales, respectively. These advances allow us to capture biologically relevant rapid diffusion of unlabeled molecules over a full range of concentrations and at length- and time-scales not accessible to other techniques.

HS-AFM line scanning detects nanoscale movements at millisecond rates, here the A5 rotation, reported by $\sim 0.3$nm height variations between the tip of the protomers and the connections in between. HS-AFM height spectroscopy (HS-AFM-HS), an approach inspired by fluorescence spectroscopy, measures height changes at microsecond rates as molecules move under the tip, here the diffusion of molecules and gives information about diffusion rates, surface concentrations, and oligomerization of unlabeled biomolecules. A small number of previous studies have also shown the potential of reducing the dimensionality of

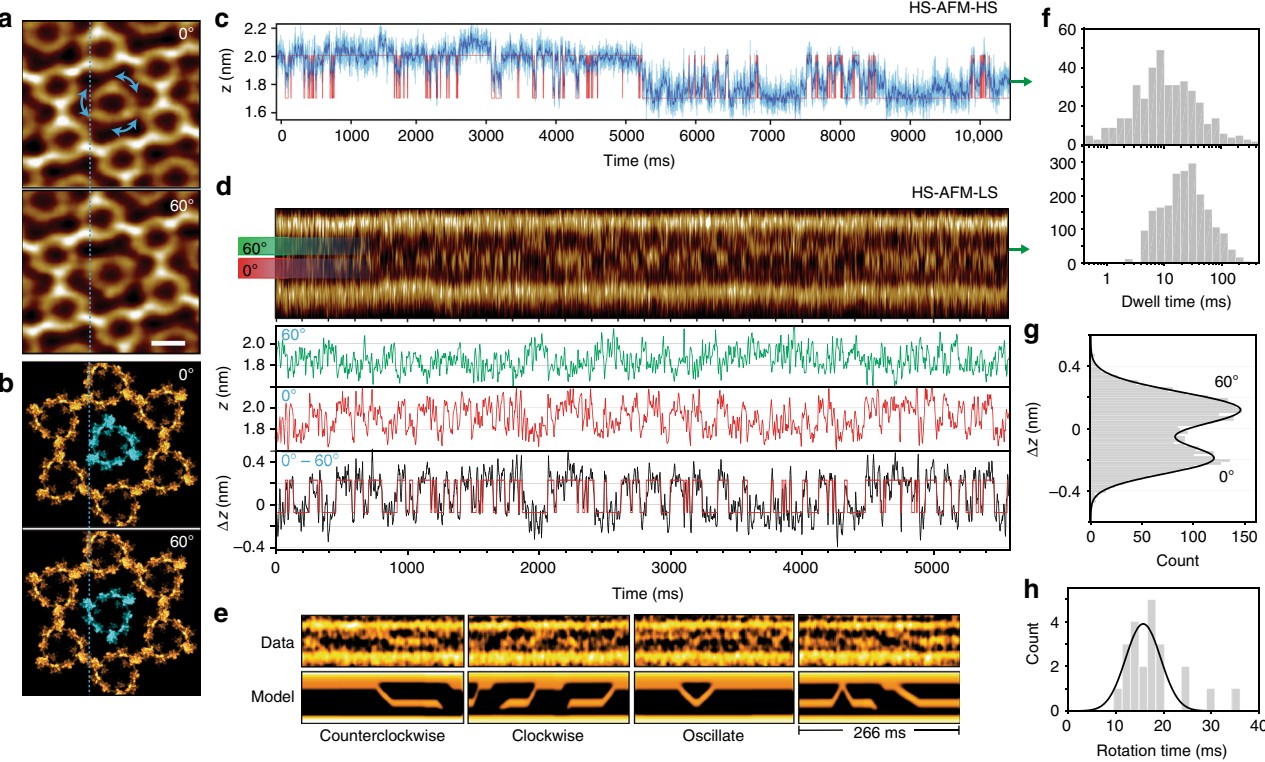

**Fig. 5** HS-AFM-HS and line scanning of A5 rotation. **a** Averaged HS-AFM images and **b** structural models, of the A5 trimer at the center of the hexagonal $p6$-lattice, captured in its two preferred orientations 0° and 60° (Supplementary Movie 1). Scale bar: 5 nm. **c** Height/time trace obtained from HS-AFM-HS measurements on one protomer of the rotating A5. Data were captured at 655 kHz (light blue) and was overlaid with filtered data over 30 points (dark blue) and an idealized two state trace (red). **d** Line scanning kymograph across one A5 protomer captured at 2.4 ms per line. Labels 60° and 0 ° (see in a) indicate the $x$-positions where the height/time traces below, in green and red respectively, were obtained. The anti-correlation of the two positions allows the height/time signal difference between the height traces at 0° and 60° to be plotted (black trace) and fitted with a two-state model (red trace). **e** Line scanning kymographs (top), and example model kymographs of rotations between the two preferred orientations (bottom). **f** Distribution of dwell-times spent in each orientation before rotation, with overlaid normalized survival plots obtained by HS-AFM-HS (upper panel) and line scanning (lower panel). Histograms each contain data from 3 different trimers each showing no significant statistical differences between molecules. **g** Histogram of the height differences obtained by subtracting height/time traces at 0° and 60° in **d**. **h** Rotation time histogram ($n = 22$) of single resolved 60° clockwise and counter-clockwise rotations in **e**

acquisition in AFM as a tool to study dynamics, however it is yet to be fully exploited[34–39].

Together; the data allows us to describe the entire annexin-V membrane association and self-assembly process in quantitative detail (Fig. 6). Initially at low $Ca^{2+}$ concentrations, single A5 diffuse on the membrane with $0.8\ \mu m^2\ s^{-1}$, upon further recruitment of A5, higher oligomers form on the membrane notably $A5_2$ and $A5_3$ that diffuse slower with $0.63\ \mu m^2\ s^{-1}$ and $0.58\ \mu m^2\ s^{-1}$, respectively. These multimeric states exist in equilibrium with each other implying interactions are reversible and weak (comparable to thermal energy). As the surface concentration increases further, crowding lowers slightly the diffusion of A5 (~10%) and $A5_2$ (~8%) and allows the formation of even higher oligomers $A5_4$ and $A5_5$ that diffuse slower than the smaller oligomers with $0.50\ \mu m^2\ s^{-1}$ and $0.46\ \mu m^2\ s^{-1}$ respectively. These higher-order oligomers are formed in a stepwise manner consistent with self-assembly models of 2D-association. As higher-order oligomers accumulate, and higher surface densities are attained, a critical 2D-concentration of ~500 molecules $\mu m^{-2}$ is reached leading to the formation of an immobile lattice. At this critical concentration when higher-order structures form, surface binding from the bulk increases significantly. This may be either a consequence of the lattice formation, which allows capture of the molecules into a structure where $k_{off}$ (in both 2D and 3D) becomes extremely low, or that the increased binding triggers

lattice formation. It seems very likely that the integration of A5 into larger-scale supramolecular structures significantly lowers the $k_{off}$ due to a form of avidity where molecules in the lattice stabilize other molecules in the lattice. In support of this, we found using HS-AFM imaging during $Ca^{2+}$ titration (addition and removal) that the lattice trimers had a different apparent $Ca^{2+}$ and membrane affinity than the non-$p6$-trimers[27]. In cellular environments, such a bias may provide a way to spatially regulate lattice formation to the membrane defect.

Here, we extend the dynamic range of HS-AFM imaging mode of ~100 ms with HS-AFM line scanning (HS-AFM-LS) to ~1 ms and with HS-AFM height spectroscopy (HS-AFM-HS) to ~10 μs. The z-sensitivity of all modes is solely limited by the precision of the detection of the cantilever deflection which is currently ~1.5Å at the bandwidth limit of the cantilever resonance frequency[40]. Using this technical toolbox, we measure rapid diffusion processes, giving access to biochemical and biophysical parameters including affinities and association/dissociation kinetics describing entirely and quantitatively the Annexin-V membrane-association process.

HS-AFM-LS and HS-AFM-HS have a wide range of applications to study microsecond dynamics of unlabeled biomolecules, such as the study of ligand-induced oligomerization of receptors and transporters, the conformational dynamics of transporters, receptors and channels during transport cycles, ligand binding and gating, respectively, or diverse enzymatic actions.

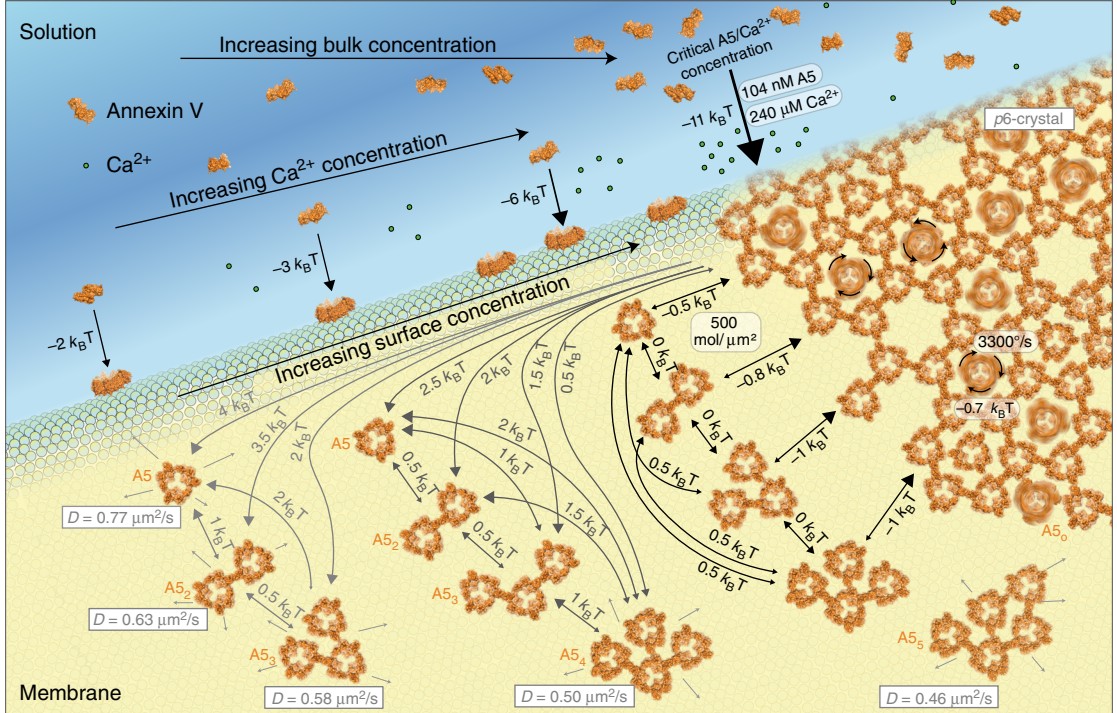

**Fig. 6** Full quantitative description of annexin-V membrane association and self-assembly. Energetic and dynamic terms of the process could be determined by HS-AFM-HS, combined with HS-AFM imaging and line scanning. The annotated illustration describes how soluble annexin-V bind to the membrane in the presence of $Ca^{2+}$ and there, as the result of 2D diffusion, association, and crowding reach a critical 2D concentration necessary for the formation of 2D-lattices that have essentially no off-rate and thus stabilize the membrane around damages

## Methods

**HS-AFM**. All AFM measurements in this study were taken by amplitude modulation mode HS-AFM (RIBM, Japan), as previously described in Miyagi et al. 2016[27]. In brief, short cantilevers (USC-F1.2-k0.15, NanoWorld, Switzerland) with spring constant of 0.15 N m$^{-1}$, resonance frequency of ~0.6 MHz and a quality factor of ~2 in buffer, were used. The HS-AFM was equipped with an illumination system allowing UV light from a mercury lamp to be focused through the same objective as the HS-AFM IR laser to release caged $Ca^{2+}$ during HS-AFM imaging or HS-AFM-HS.

**HS-AFM-HS**. HS-AFM-HS measurements were taken directly after HS-AFM imaging by stopping the $x$–$y$ piezos, leaving the tip at the center of the previous image with the z-feedback remaining active. Measurements were taken with a free oscillation amplitude of ~3 nm and a set-point amplitude at >90% of the free amplitude. Feedback settings were optimized to maximize feedback response speed. Z-piezo data was captured with home written software and a data acquisition board with a maximum acquisition rate of 2,000,000 samples s$^{-1}$ (LabView programming, NI-USB-6366 card, National Instruments, USA).

**Sample preparation**. The annexin-V used in this study was purchased from Sigma-Aldrich (Annexin-V, 33 kD from human placenta) and all lipids (dioleoyl-phosphatidyl-choline (DOPC) and dioleoyl-phosphatidyl-serine (DOPS)) from Avanti polar lipids. Annexin binding and crystallization on SLBs was achieved by addition of annexin to a preformed lipid bilayer. In brief, lipids were solubilized in chloroform at a ratio of DOPC:DOPS = 8:2. The solvent solubilized mixed lipids were dried by a nitrogen flow and further dried in a vacuum chamber for 2 h. Then the dried lipid was resuspended into a buffer solution containing 10 mM HEPES at pH 7.4, 150 mM NaCl and 2 mM $CaCl_2$ to form multilamellar vesicles. As the final step in lipid preparation the suspension was tip-sonicated for 10 min to obtain small unilamellar vesicles (SUVs). 1.5 μl of the SUV solution with a total lipid concentration of 0.1 mg ml$^{-1}$ was deposited onto freshly cleaved mica to form SLBs through vesicle fusion. The excess lipids, after SLB formation, were rinsed first with deionized water followed by buffer. A5 was added to the imaging solution at varying volumes to achieve desired bulk concentrations with an observation buffer 10 mM HEPES at pH 7.4, 150 mM NaCl with $CaCl_2$ ranging between 0 and 2 mM.

**$Ca^{2+}$ uncaging experiments**. In the $Ca^{2+}$ uncaging experiments, the observation buffer contained 10 mM HEPES at pH 7.4, 150 mM NaCl, 1 mM $CaCl_2$ and 1.25 mM o-nitrophenyl EGTA tetra-potassium salt (NP-EGTA). The ensemble of 1 mM $CaCl_2$ and 1.25 mM NP-EGTA forms 1 mM caged $Ca^{2+}$, the slight excess of NP-

EGTA assures complete $Ca^{2+}$-chelation. NP-EGTA has a high selectivity for $Ca^{2+}$ upon UV illumination, its $Ca^{2+}$ dissociation constant increases 12,500-fold from 80 nM to >1 mM. During uncaging HS-AFM-HS measurements, UV light from the mercury lamp was allowed to pass through to the AFM-scanning area using a shutter and aperture to control the intensity. The diameter of the UV spot size was around 1 mm$^2$ including the AFM scanning area.

**Data analysis**. The HS-AFM movies were drift corrected and contrast adjusted by a laboratory-built image analysis software in ImageJ. To minimize $x$–$y$ drift during HS-AFM-LS and HS-AFM-HS, the scanner and tip holder were made sure to be in stable positions and large $x$–$y$ translations were avoided directly before HS-AFM-HS/HS-AFM-LS data capture. Additionally, to ensure the HS-AFM tip is on the same molecule for a certain period and no significant changes in tip radius has occurred, HS-AFM image sets were captured directly before and after HS-AFM-HS measurements, allowing the total drift to be measured and tip quality to be assessed. The HS-AFM line-scanning kymographs were contrast adjusted and assembled by routines and self-written analysis software in ImageJ. HS-AFM-HS and HS-AFM-LS height/time traces were analyzed using self-written routines in MATLAB (Matlab, Mathworks, Natick, MA, USA).

**Code availability**. MATLAB codes used for analysis are available from the corresponding author on reasonable request.

**Reporting Summary**. Further information on experimental design is available in the Nature Research Reporting Summary linked to this article.

## Data Availability

The data sets generated and/or analyzed during the current study are available from the corresponding author on reasonable request.

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

## Acknowledgements

We thank Yi-Chih Lin and Atsushi Miyagi for technical and topical support.

## Author contributions

S.S and G.R.H conceived and designed the experiments. G.R.H performed the experiments. S.S and G.R.H analyzed the data. S.S and G.R.H wrote the paper.

## Additional information

**Competing interests:** The authors declare no competing interests.

