## [Peer Review File · Nature Communications]

Reviewers' comments:

Reviewer #1 (Remarks to the Author):

In the manuscript, Heath and Scheuring utilized high-speed AFM as line- and height-profile tools with an angstrom-order spatial resolution and unprecedented temporal resolution of a few ms and ~20 us, respectively. The usefulness of the method was demonstrated by investigating dynamic features of a membrane associated protein, annexin V. The experimental approach is very simple, but the method presented here may grow as one of the useful biophysical methods to obtain important physical parameters of unlabeled biomolecules on a surface.

However, technical details should be shown more clearly. In addition, there are many ambiguous points in the current version, which may mislead the readers. Moreover, some additional information would enhance the interest, reliability and impact of the manuscript. I list below specific comments that should be clarified before publication.

Major concerns:

1. It is admirable that the authors dramatically improved the temporal resolution of height-spectroscopy based on AFM. However, the temporal resolution is still more than ~20 us. Thus, "microsecond dynamics" stated in the current title should be toned down accordingly. Also, it is better to add "on a surface" in the title.
2. Fig. 1e, Fig. 5d and Supplementary Fig. 1 are the same. Also, Fig.1f and Fig.5f bottom are the same. They are very awkward. Order of describing the paper should be well reconsidered.
3. The words of Annexin-V and A5 are randomly used. The authors should use the word with a certain rule.
4. There is no doubt in estimating the surface concentration and dynamics of freely moving molecules by the height spectroscopy. This is because the mechanical drift of AFM may be negligible small compared with the diffusional motions of biomolecules placed on a lipid membrane in the liquid phase. However, there is doubt in height spectroscopy on a specific point of a biomolecule (e.g., Fig. 5c). In fact, the mechanical drift of AFM can be seen in Fig. 1b and Fig. 4a, b (See the position shift of the holes seen at the bottom right and upper right). How do the authors ensure that the AFM tip is on the same point of the molecule for a certain period? In other word, how long can the authors perform the HS-AFM-HS at the certain position on a protein without worrying about the mechanical drift of AFM? More simply, what is a position accuracy of HS-AFM-HS? Also, when the authors performed the height spectroscopy on immobile points of the hexagonal p6-lattice, the authors could detect the expected height profile? These things should be the key issue to evaluate the performance and usefulness of HS-AFM-HS presented here. The authors should describe these things with appropriate additional data. Otherwise, the results shown in Figs. 5c and 5f top, and related discussions have no meaning.
5. How do the authors determine the threshold height in the height spectroscopy? This value is always the same or different? Moreover, the authors should use a different mark from "T" for the threshold height, because "T" also represents the absolute temperature. How about "H_T"?
6. Line showing the threshold height should be drawn on the all height spectroscopy data.
7. How do the authors measure the pulse width? This should be clearly described in the manuscript. Typically, for Fig. 3b-d and Supplementary Fig. S2, it is very hard to determine the width.
8. L102, L165, L424 and L463: Judging from Supplementary Fig. S2, the temporal resolution is more than ~20 us.

9. L280: From Fig. 3e, how do the authors estimate the critical surface coverage concentration of Ca^{2+} ?

10. How do the authors construct Fig. 3g? More detail explanation should be added.

11. L405: the dwell time values of 18.6 ± 0.5 ms and 32.0 ± 0.3 ms determined by HS-AFM-HS and HS-AFM-LS were wrongly cited in Fig. 5f where the values are 21 ms and 30 ms, respectively. Which is correct? Also, these values were only obtained from one molecule and one experiment? Then, can the authors safely state that these values are statistically reliable to characterize the molecular features? The authors should confirm that these values are mostly consistent with those obtained on the different molecules. Furthermore, the reviewer feels that this small amount of statistics leads to an unclear discussion of L415-420.

12. The histogram of Fig. 5f top was not fitted by the single exponential decay at all. This indicates the observed phenomena can be described another reaction model (e.g., double exponential decay)? Also, the fitting seen in Fig. 5f bottom is not so nice.

13. A large discrepancy between the data and model can be seen in Fig. 5e. How do the authors generate the model? Also, how do the authors evaluate the credibility of the model?

14. The reviewer feels that Fig. 5h is statistically poor. More data should be accumulated to estimate the rotation time more accurately.

15. L465: The authors stated that the bandwidth of HS-AFM-HS is limited by the cantilever resonance frequency. However, this is not entirely correct. To be exact, the bandwidth of HS-AFM-HS is determined by the response time of the all devices included in the z-height feedback loop. More practically, the bandwidth can be quantified by the response times of the z-scanner and the cantilever (the slowest devices in the feedback loop). Thus, the response time of the z-scanner is another key parameter in the manuscript. The authors should describe this value somewhere.

16. Many values estimated in the manuscript have no error values.

17. What is the minimum size of molecule that can be investigated by the current HS-AFM-HS? This would be important information for the general reader.

18. Supplementary Figure S5: What is the color code seen in each graph? It seems that the fitting curves do not fit data at all. The values estimated from these fittings here are reliable?

Minor concerns:

1. The abstract exceeds 150 words extensively.

2. L53: "Whilst able to provide...", there is no subject in this sentence.

3. L137: "tip parachuting", brief explanation is necessary for inexpert readers, or refer literatures.

4. Fig. 1i: Data at high temporal resolution regime (e.g., 0 – 10 ms) should be also shown to demonstrate the wide variety of height profile detected.

5. Fig. 1d, g: The values beside the images should include the temporal range (e.g., a few ms and ~20 us, respectively) as shown in Fig. 1a.

6. The multiple fitting seen in Fig. 1f is wrong? Although the graph is exactly the same as Fig. 5f bottom, the latter is fitted by the single exponential decay. Or, either is wrong?

7. Cite a literature for Eq.1. Also, τ_D should be used in the main text.

8. L194: It is valuable to summarize the mean value and estimation error of r_p for each A5 oligomer in a table.
9. L202: r is r_p ? Also, explain $t_z > T$ and t_{total} in the main text.
10. L228: It is more kind if the authors add the information of the A5 concentration here.
11. Technically, AFM imaging and height spectroscopy cannot be performed at the same time. In this sense, the blue lines from Fig. 4a, b should be drawn outside of the panel of Fig. 4c or removed from the figures.
12. Typically, the A5 height is detected as 1.0-1.5 nm in HS-AFM-HS. Why the height seen after ~ 85 s in Fig.4c are so high (more than 2 nm)?
13. Fig. 4c: Showing typical three data at high temporal resolution regime (e.g., 20 – 30 ms, 60-70 ms and 80 – 90 ms) are valuable to demonstrate the height profile change over the time and to directly link the results of Fig. d, e.
14. k_B and K_b are randomly used. They should be used uniformly.
15. The blue arrows seen in Fig. 5a should be bidirectional because the A5 rotates both clockwise and counter-clockwise.
16. L504: "HEPES NaCl" is "HEPES NaOH" correctly?

Reviewer #2 (Remarks to the Author):

In their manuscript „High-Speed AFM Height Spectroscopy (HS-AFM-HS): Microsecond dynamics of unlabeled biomolecules”, Heath and Scheuring present a methodology for probing diffusional motion of single molecules and their surface concentration at a solid-liquid interface. The authors employ this methodology to quantify the kinetic constants that govern the 2D self-assembly of the protein annexin-V, which is proposed to play a role in harnessing damaged membranes.

The methodology uses a high-speed atomic force microscope that can acquire topographic images of a 2D surface in the ~ 100 ms regime. To improve temporal resolution, the authors reduce the dimensionality of the system. Continuous scanning of a single line, coined “high speed AFM line-scanning”, improves the temporal resolution by two orders of magnitude (~ 1 ms). Further reduction of the dimensionality by probing the height at a single (x,y)-position yields a temporal resolution that is limited by the z-piezo response time, or ~ 20 μ s.

Whereas the concept of reducing the dimensionality to increase the temporal resolution of AFM is not new, this is (to my knowledge) the first example that employs a high-speed atomic force microscope, thereby enabling unprecedented time-resolution. In turn, new applications in biology and possibly in interface science in general might be addressed. While I have some detailed comments that could further improve this work (see below) I find the manuscript well-written and generally interesting.

Detailed comments

-The term “height spectroscopy” implies that a spectrum of heights can be addressed, and that different heights provide additional information. However, the height information is solely used to address the presence or absence of diffusing proteins under the scanning probe, using a threshold value. How accurate is the measurement of height, and how does this depend on the time it is measured? More specifically, in supplementary figure S2, step-functions with $z = 2$ nm input show decreasing measured heights for reduced dwell times. Can the authors explain this? How does the method perform for different molecular heights, in particular for smaller molecules? It would be

insightful to perform experiments wherein the z-piezo is moved to different heights (<2 nm), and for different dwell times, to identify the dependence of the temporal resolution on the height of diffusing molecules.

- The method quantifies dwell times and frequencies of molecules moving under the tip. Other parameters are determined indirectly and are subject to error propagation or fitting quality. The authors should provide appropriate statistical analysis (reduced chi-square of fits, confidence intervals for the fit parameters, ...) and indicate the meaning of the errors on values where appropriate.

- Supported lipid bilayers as a substrate are less dynamic than "free-standing" bilayers. The authors should comment on how this would affect the dynamics of annexin-V assembly, and to what extent the kinetics can be translated to a cellular context.

- Drift correction is performed using a "laboratory built image analysis software". How does this software deal with drift, and (how) is it applied to HS-AFM-LS and HS-AFM-HS?

- Tip size can change during the measurement. How does tip size affect the measured diffusion times of annexin-V?

Reviewer #3 (Remarks to the Author):

The authors developed high-speed AFM height spectroscopy for directly detecting the motions of molecules under a HS-AFM tip at a fixed position. They measured simultaneously surface concentrations, diffusion coefficients and oligomer sizes of annexin-V on model membranes and described the entire annexin-V membrane-association and self-assembly process in great detail and quantitatively by using HS-AFM-HS and HS-AFM imaging. Their method has Angstrom spatial and microsecond temporal resolutions of unlabeled molecular fluctuations. In my opinion, this work is interesting and could be published in Nature Communication. However, before publication, I would like to propose a revision on the following points.

1 Did different ratios of DOPC:DOPS affect the annexin-V membrane-association and self-assembly process?

2 The authors determined the diffusion coefficient of dimers and trimer of A5. Can they estimate their theoretical values? Can they compare their results with other methods?

3 The authors mentioned that "the A5 p6-lattice assembled, it is made up of two-thirds p6-trimers that constitute the 'honeycomb lattice' and one-third non-p6 trimers that are not strictly part of the lattice being trimers sitting on the 6-fold symmetry axis". Why?

4 Can this method be used for monitoring the annexin-V membrane-association and self-assembly process on the surface of cell?

RED = comments to reviewer

GREEN = Changes made in text

BLUE = extracts from text

Reviewer #1 (Remarks to the Author):

In the manuscript, Heath and Scheuring utilized high-speed AFM as line- and height-profile tools with an angstrom-order spatial resolution and unprecedented temporal resolution of a few ms and ~20 μ s, respectively. The usefulness of the method was demonstrated by investigating dynamic features of a membrane associated protein, annexin V. The experimental approach is very simple, but the method presented here may grow as one of the useful biophysical methods to obtain important physical parameters of unlabeled biomolecules on a surface. However, technical details should be shown more clearly. In addition, there are many ambiguous points in the current version, which may mislead the readers. Moreover, some additional information would enhance the interest, reliability and impact of the manuscript. I list below specific comments that should be clarified before publication.

Major concerns:

1. It is admirable that the authors dramatically improved the temporal resolution of height-spectroscopy based on AFM. However, the temporal resolution is still more than ~20 μ s. Thus, “microsecond dynamics” stated in the current title should be toned down accordingly. Also, it is better to add “on a surface” in the title.

We thank the reviewer for the overall positive (admirable) evaluation of our work. Indeed, considering that HS-AFM typically takes movies at 100ms to 1000ms frame acquisition speed, we gain about 4 orders of magnitude.

We disagree however with the reviewer that “on a surface” should be specified in the title. Whilst we do perform the experiments on a surface the use of AFM in the title indicates that we are on a surface. Additionally, we feel the extra text would overextend the title making it less appealing to the general reader.

Furthermore, we would like to point out that ~20 μ s is a rather conservative estimate (based on Supplementary Figure 2). In this work, for the object under analysis, we didn't have to push the limits further. As the reviewer certainly agrees, faster events that are not followed by the feedback, are recorded in the amplitude error signal (where the HS-AFM cantilever oscillates at 500kHz (2 μ s)).

Here we introduce a novel HS-AFM modality that is still improvable. To tone down the message, we change the title from “microsecond dynamics” to “ μ s-dynamics” indicating that we are reaching the μ s-range and not 1 microsecond.

2. Fig. 1e, Fig. 5d and Supplementary Fig. 1 are the same. Also, Fig. 1f and Fig. 5f bottom are the same. They are very awkward. Order of describing the paper should be well reconsidered.

To avoid repetition, **Figure S1** and **Fig. 1e** have been changed. Also **Fig. 5f** has been altered to a histogram with a log x-scale to allow the reader to make easier comparison between HS-AFM-LS and HS-AFM-HS.

We disagree with the reviewer that the paper order should be revised. Currently the order of describing the results is as follows:

1. Introduction of how reduction of dimensionality of data acquisition improves temporal resolution to introduce line scanning and height spectroscopy (Fig. 1). This gives a concise overview and allows the reader to quickly grasp the technique progression.
2. Annexin binding, 2D association and lattice formation as measured by height spectroscopy, thus describing the major development and focus of the manuscript.
3. Annexin dynamics in the 2D lattice as measured by line scanning and height spectroscopy providing direction comparison between the techniques.

We find this the most logical ordering of the manuscript, with a focus on the height spectroscopy element from the outset. An alternate possible order would move figure 5 to figure 2, however we feel this would bring too much focus on line scanning and works backwards in terms of the crystallization process.

Finally, the advantage of an online-only journal like Nature Communication that puts little pressure on manuscript length should allow us to hold the introductory figure 1 that exemplifies the concept.

3. The words of Annexin-V and A5 are randomly used. The authors should use the word with a certain rule.

We do describe a rule for this terminology in L226, Annexin-V relates to a single annexin protein (the species) existing primarily in solution whilst A5 relates to the trimeric form found on membranes. Then we term $A5_2$ (dimer of the membrane-bound trimer), $A5_3$ (trimer of the membrane-bound trimer), etc, when higher oligomers of the membrane-bound trimer are described.

We apologize that this did not come across and agree with the reviewer that the current introduction at L226 could cause confusion. To make it clearer we have introduced this rule sooner in the manuscript:

L124:

However, 2D-scanning is not able to resolve the highly mobile membrane-bound annexin trimers (A5) during the early stages of the assembly process, as they diffuse too quickly to be resolved when images are acquired at frame rates of $1s^{-1}$ to $10s^{-1}$; instead, only streaks in the fast scanning direction (x) are observed (herein A5 is used to refer to the membrane-bound trimeric form of annexin-V).

Also, any misuses of A5 or annexin-V from this rule have been corrected throughout the manuscript.

4. There is no doubt in estimating the surface concentration and dynamics of freely moving molecules by the height spectroscopy. This is because the mechanical drift of AFM may be negligible small compared with the diffusional motions of biomolecules placed on a lipid membrane in the liquid phase. However, there is doubt in height spectroscopy on a specific point of a biomolecule (e.g., Fig. 5c). In fact, the mechanical drift of AFM can be seen in Fig. 1b and Fig. 4a, b (See the position shift of the holes seen at the bottom right and upper right). How do the authors ensure that the AFM tip is on the same point of the molecule for a certain period? In other word, how long can the authors perform the HS-AFM-HS at the certain position on a protein without worrying about the mechanical drift of AFM? More simply, what is a position accuracy of HS-AFM-HS? Also, when the authors performed the height spectroscopy on immobile points of the hexagonal p6-lattice, the authors could detect the expected height profile? These things should be the key issue to evaluate the performance and usefulness of HS-AFM-HS presented here. The authors should describe these things with appropriate

additional data. Otherwise, the results shown in Figs. 5c and 5f top, and related discussions have no meaning.

The reviewer makes an important point, from HS-AFM imaging we know there is mechanical drift, i.e. lateral drift in x-y directions. The scanner stage can drift by some nanometers in x- and y-dimension, especially during the seconds after execution of a position or scan range change due to piezo-relaxation (Thomson et al., 1996). To assess the mechanical drift, we can capture HS-AFM images of the annexin lattice for several minutes and then use image correlation alignment software to find the x-y translations required to align the image set. Under normal imaging conditions the total drift distance was found to vary from as low as 0.02 nm/s with a well equilibrated system (as shown in supplementary Fig. 8) up to 0.1nm/s. This drift is relatively slow 10-50s/nm in comparison to the tip radius and the area of interest, which is an additional advantage of HS-AFM-HS over fluorescence techniques where bleaching often limits the total time a molecule can be observed with high temporal resolution.

Furthermore, HS-AFM-HS is used when monitoring fast dynamic events beyond the speed of acquisition in imaging mode (much faster than 10s-1), thus even for the cases when the drift is relatively strong, one still captures hundreds of events on 1nm within 10s. If one is to analyze slower dynamics, then line scanning or imaging mode is appropriate and drift can be corrected based on the recorded environmental features in either one dimension (line scanning) or two dimensions (imaging).

To minimize x-y drift, the scanner and tip holder were placed in stable positions and large x-y translations were avoided directly before HS-AFM-HS capture. Additionally, to ensure the AFM tip is on the same molecule for a certain period, HS-AFM images were taken directly before and after HS-AFM-HS measurements were captured, allowing the total drift to be measured.

Performing HS-AFM-HS on immobile trimers in the hexagonal p6-lattice produced the expected height profile, this appears simply as background noise with approximately the same RMS as on the surface of the membrane in the presence of no proteins.

The following supplementary figure and movie S2 have been added to the supporting information:

Supplementary Figure 8) Quantification of lateral drift. a) x-y position change during HS-AFM imaging as determined by image correlation alignment with subpixel interpolation of supporting movie S2. The translations are overlaid onto an annexin trimer (A5) to illustrate the scale of lateral drift during HS-AFM-HS measurements. b) Total drift distance from initial position over time with linear fit to determine a drift rate of 51s/nm (or 0.0195 nm/s). It should be noted that

alignment was performed on images with 0.5nm/pix and thus apparent short timescale drift noise is expected to be due to alignment accuracy.

The following text has also been added to the main text of the manuscript:

L394:

Performing HS-AFM-HS on immobile trimers in the hexagonal p6-lattice produced a height trace with only one state with the expected background noise of $\pm 0.12\text{nm}$. It should be noted the scanner stage can drift by some nanometers in x- and y-dimension, especially during the seconds after execution of a position or scan range change due to piezo-relaxation. To assess the mechanical drift, we can capture HS-AFM images of the A5-lattice for several minutes and then use image correlation alignment software to find the x-y translations required to align the image set. Under normal imaging conditions the total drift distance was found to vary from as low as 0.02 nm/s with a well-equilibrated system (as shown in Supplementary Fig. 8) up to 0.1nm/s. This drift is relatively slow 10-50s/nm in comparison to the tip radius, the area of interest and the biological dynamics to be analyzed ($\gg 10\text{s}^{-1}$) and suggests that HS-AFM-HS can be positionally accurate for 10s of seconds. Such stability provides an additional advantage of HS-AFM-HS and HS-AFM-LS over fluorescence techniques where bleaching often limits the total time a molecule can be observed with high temporal resolution.

The following text has been added to methods:

L552:

To minimize x-y drift during HS-AFM-LS and HS-AFM-HS, the scanner and tip holder were made sure to be in stable positions and large x-y translations were avoided directly before HS-AFM-HS/HS-AFM-LS data capture. Additionally, to ensure the AFM tip is on the same molecule for a certain period and no significant changes in tip radius has occurred, HS-AFM image sets were captured directly before and after HS-AFM-HS measurements, allowing the total drift to be measured and tip quality to be assessed.

5. How do the authors determine the threshold height in the height spectroscopy? This value is always the same or different? Moreover, the authors should use a different mark from “T” for the threshold height, because “T” also represents the absolute temperature. How about “H_T”?

This value depends on the noise level of each measurement, which only changes very slightly leading to H_T values of $0.8 \pm 0.1\text{nm}$. We agree with the reviewer that “T” may cause confusion and therefore have changed T to H_T throughout including in Fig. 1.

The selection of a threshold height is described in line 203:

“The threshold height T was not an arbitrary value, but chosen based on the background noise level of the height trace, significantly far away from the noise distribution at 5σ so that the probability of mistaking diffusion events is 0.00006%.”

To further clarify we have updated the text as follows:

L207:

The threshold height H_T was not an arbitrary value but chosen based on the background noise level of the height trace, significantly far away from the noise distribution at 5σ so that the probability of mistaking diffusion events is 0.00006% (typically this corresponds to $H_T = 0.8 \pm 0.1\text{nm}$).

6. Line showing the threshold height should be drawn on the all height spectroscopy data.

We added such a H_T line in Figure 1i (where we introduce the height spectroscopy) but avoid this extra annotation in the following figures as we think it is not needed as H_T is quite constant and described in the text (further detailed in revision). Also, we prefer not to clutter the figures and raw data within. The annotation on Figure 1i has been made more visible.

7. How do the authors measure the pulse width? This should be clearly described in the manuscript. Typically, for Fig. 3b-d and Supplementary Fig. S2, it is very hard to determine the width.

The pulse width is taken as the time a molecule spends under the tip with height above the threshold height H_T . We agree with the reviewer that this detail was missing and should be included in the manuscript, therefore the following has been added:

L174:

Measuring the time duration of each peak above H_T gives a distribution of dwell times corresponding to the range of times molecules spend under the tip (Fig. 1j), with the fastest events being only about 20 μ s long.

8. L102, L165, L424 and L463: Judging from Supplementary Fig. S2, the temporal resolution is more than ~20 us.

Supplementary Fig. 2 gives a very conservative estimate of the temporal resolution, because we send merely square pulses; this never occurs in a real experiment. We have done it to assess how precise the dwell time is reported by the HS-AFM. Typically to assess the reaction bandwidth of an AFM, a sine wave of higher and higher frequency is sent and the response (until resonance occurs) is monitored. From these measurements, we (and others) have found that HS-AFM has ~100kHz bandwidth (~10 μ s), and indeed in the real experimental data, we detect dwell times of ~10 μ s (Figure 3a). Anyway, Supplementary Fig. 2 shows pulses of different dwell times are reliably reported down to 20 μ s.

L170:

For future applications of even faster events, the amplitude damping of the cantilever oscillation can be monitored, which should report about events beyond the feedback bandwidth.

9. L280: From Fig. 3e, how do the authors estimate the critical surface coverage concentration of Ca^{2+} ?

The critical concentration of Ca^{2+} at which A5 reaches 500 μm^{-2} is calculated from the exponential fit (red line in Figure 3e) of the A5 surface concentration vs bulk Ca^{2+} concentration (note, the y-axis is in log-scale). The Ca^{2+} concentration at which the critical A5 surface concentration (500 μm^{-2}) is reached is defined as the critical Ca^{2+} (kink in the dwell time distribution vs surface concentration in Fig. 3g). Our writing was misleading, as this critical A5 surface concentration is not discussed until later in the main text and therefore may not be clear. To clarify for the reader, additional explanation in the text describing this has been moved and updated as follows:

L314:

For experiments in varying Ca^{2+} concentrations, fits to the A5 surface concentration vs. Ca^{2+} bulk concentration (Fig. 3e) show this critical A5 surface coverage is reached at $240 \pm 10 \mu M$ Ca^{2+} , in agreement with previous work on model membranes and cells.^{26,33}

10. How do the authors construct Fig. 3g? More detail explanation should be added.

Fig. 3g is constructed using the average of all dwell times measured over a 1 second period and plotted against the measured surface concentration within that 1 second time window. This information is provided in the figure caption after the descriptions of parts g), h) and i):

“Dwell-times in (g), oligomer distributions in (h) and surface concentrations (x-axes in h and g) are averages over 1-second time-windows during the height spectroscopy analysis.”

To clarify and give more detailed explanation the text has been updated to read:

Fig. 3... g) Graph displaying the average A5 diffusion dwell-times under the tip as a function of surface concentration. Average dwell-times are calculated from dwell-times observed within 1-second time-windows. At ~ 500 molecules per μm^2 surface density (dashed line), the average dwell-times changes regime indicative of the formation of higher order oligomers. h) Experimental and i) theoretical oligomer distributions as a function of surface concentration. Oligomer distributions in (h) and surface concentrations (x-axes in h and g) are calculated over 1-second time-windows during the height spectroscopy analysis.

11. L405: the dwell time values of 18.6 ± 0.5 ms and 32.0 ± 0.3 ms determined by HS-AFM-HS and HS-AFM-LS were wrongly cited in Fig. 5f where the values are 21 ms and 30 ms, respectively. Which is correct? Also, these values were only obtained from one molecule and one experiment? Then, can the authors safely state that these values are statistically reliable to characterize the molecular features? The authors should confirm that these values are mostly consistent with those obtained on the different molecules. Furthermore, the reviewer feels that this small amount of statistics leads to an unclear discussion of L415-420.

We apologize for this confusion, the values stated in the figure (21ms and 30ms) are from the decay constants obtained by exponential fits to the lifetime plots (see Q12) whereas the values in the text are averages of one trimer. To clarify and in response to Q12, Fig. 5 has been amended, removing the overlaid lifetime plots and exponential fitted values. The histograms in Fig. 5f have now been plotted with a log x-scale to allow the reader to make easier comparison between HS-AFM-LS and HS-AFM-HS. The rotation dwell times obtained by height spectroscopy and line scanning are statistically reliable, the values were obtained from 6 different trimers comprising a total of 2454 transitions. All transition from different trimers have now been included in the mean and error analysis. To reflect these points the figure caption and main text has been updated as follows:

Fig. 5... f) Distribution of dwell-times spent in each orientation before rotation obtained by HS-AFM-HS (upper panel) and line scanning (lower panel). Histograms each contain data from 3 different trimers each showing no significant statistical differences between molecules.

L442:

The movements between the 0° and 60° states, detected by HS-AFM-HS and line scanning result in average dwell-times of 26 ± 3 ms (s.e.m., $n = 401$) and 35.0 ± 1 ms (s.e.m., $n = 2053$), respectively (**Fig. 5f**).

12. The histogram of Fig.5f top was not fitted by the single exponential decay at all. This indicates the observed phenomena can be described another reaction model (e.g., double exponential decay)? Also, the fitting seen in Fig. 5f bottom is not so nice.

As described in the figure caption these lines were not exponential fits but lifetime plots:

“f) Distribution of dwell times spent in each orientation before rotation, with overlaid normalized survival plots obtained by HS-AFM-HS (upper panel) and line scanning (lower panel).”

Fig. 5f has been replotted to avoid misunderstanding (see also response to Q11), and these lifetime plot lines are not there anymore.

13. A large discrepancy between the data and model can be seen in Fig. 5e. How do the authors generate the model? Also, how do the authors evaluate the credibility of the model?

Although showing similarity, the models in Fig 5e are not fitted to the data, these images show example sets of model transitions for the readers comparison. The features (eg slopes interconnecting the two states) in the data and model corresponding to molecular rotations are easily recognizable.

A more detailed description of the model is given in Fig S1:

Supplementary Figure S1) Model Line scanning measurements of the rotation of A5 at the 6-fold symmetry axis in the *p6*-lattice. Line scanning kymograph (top), and model line kymograph (middle). Model kymographs were created by rotating a structural model of the annexin lattice (after convolution to mimic a 1nm AFM tip radius) between the two preferred orientations (images labelled 60° and 0°). The model kymographs were created by plotting the profile indicated by the blue arrows over time during several random 60° rotations with the trimer remaining stationary between rotations. Images and labels (bottom), 60 or -60°, indicate the positions where the model trimer undergoes either a clockwise or counter-clockwise rotations.

To clarify, additional reference to Fig. S1 has been added in main text when discussing Fig. 5e:

L434:

...the characteristics of these transitions depend on the initial state and the direction of rotation (supplementary Fig. 1).

Also, the caption of Fig. 5 has been updated as follows:

Fig. 5... Line scanning kymographs (top), and example model kymographs of rotations between the two preferred orientations (bottom).

14. The reviewer feels that Fig. 5h is statistically poor. More data should be accumulated to estimate the rotation time more accurately.

Although appearing statistically poor when plotted as a histogram, the estimate for the rotational velocity is based on 22 measurements. The Gaussian fit has an R-squared value of 0.62 and a Lilliefors test gives >95% confidence that these values come from a normal distribution.

The caption of Fig 5 has been updated to give the number of measurements.

Fig. 5...

...h) Rotation time histogram ($n=22$) of single resolved 60° clockwise and counter-clockwise rotations in (e).

15. L465: The authors stated that the bandwidth of HS-AFM-HS is limited by the cantilever resonance frequency. However, this is not entirely correct. To be exact, the bandwidth of HS-AFM-HS is determined by the response time of the all devices included in the z-height feedback loop. More practically, the bandwidth can be quantified by the response times of the z-scanner and the cantilever (the slowest devices in the feedback loop). Thus, the response time of the z-scanner is another key parameter in the manuscript. The authors should describe this value somewhere.

We agree with the reviewer that the bandwidth of HS-AFM-HS is determined by total response time of all devices included in the z-height feedback loop.

In brief, while the cantilever used here oscillates at 625kHz, the z-scanner resonance frequency is at about 150kHz, the feedback loop when measured to follow a sine wave about 100kHz, and the response to square pulses about 50kHz (Fig. S2).

Thus, the reviewer is right, the entire feedback loop is currently the limiting factor as described throughout the manuscript and directly tested in Fig S2, with the square pulses. On L465 we are stating that the precision of the detection of the cantilever deflection is currently $\sim 1.5\text{\AA}$ at the bandwidth of the cantilever resonance frequency. This statement hints towards the potential for even greater temporal resolution HS-AFM-HS using only the deflection signal, and/or anticipating further developments where feedback operation reaches the bandwidth of the cantilever.

To give the reader more information on limits and developments HS-AFM bandwidth we have added reference to the following recent publication on this matter:

L668:

40. Miyagi, A. & Scheuring, S. A novel phase-shift-based amplitude detector for a high-speed atomic force microscope. *Rev. Sci. Instrum.* **89**, 083704 (2018)

16. Many values estimated in the manuscript have no error values.

We agree with the reviewer, apologize for the omitting, and have now added errors values throughout the manuscript. Standard deviation values for the dwell time fittings in Fig. 2 and surface concentrations have been added to the main text. Also, the propagated error values have been added to the D values discussed in the main text. Error values based on the 95% confidence intervals of the fitting of equation 5 to the data have also been added to the K_D values in table 1 and fits in Fig. S5. Meanings of errors have also been added where an error of a value is mentioned for the first time in the text.

17. What is the minimum size of molecule that can be investigated by the current HS-AFM-HS? This would be important information for the general reader.

In the z direction, a molecule can be detected if there is a sufficient height which can be detected above the background noise ($>0.1-0.2\text{ nm}$), which is likely the case for almost all membrane- and membrane-associated- proteins. For the x-y dimension the limit is diffusion dependent, a molecule can be detected as long as the diffusion rate is slow enough to be detected within $10-20\mu\text{s}$. This generates a diffusion dependent minimum lateral size according to the following relation:

$$w = 2(\tau_{min}D)^{0.5}$$

For a molecule diffusing at $1\mu\text{m}^2/\text{s}$ this corresponds to a width of $6.6 \pm 1.3\text{nm}$ assuming a temporal resolution of $10-20\mu\text{s}$. To display these limits a novel supporting figure has been added to the supplementary with reference to it in the main text.

Inversely, a dwell-time of $33\mu\text{s}$ for a single annexin-V protomer would imply an unrealistically slow diffusion coefficient of $0.08\mu\text{m}^2/\text{s}$ (for the full molecular diffusion/size range currently accessible by HS-AFM-HS see Supplementary Fig. 3).

Supplementary Figure 3) Diffusion-dependent lateral molecule dimensions accessible by HS-AFM-HS. Red region classifies molecule sizes/diffusion rates which would diffuse under the tip with nominal dwell-times $< 10\mu\text{s}$ and therefore would generally be undetectable. The green region classifies detectable molecules (plot assumes a AFM tip radius of 1nm).

18. Supplementary Figure S5: What is the color code seen in each graph? It seems that the fitting curves do not fit data at all. The values estimated from these fittings here are reliable?

The color code in each graph is data based on data point density. To clarify these plots, we have amended the entire Fig S5, the data has been binned in the x-axis to give averaged values with standard deviation error bars. This updated format of plotting also allows the fits to be more easily assessed. To further show the reliability of the fits, 95% confidence interval fits have been added to all plots in Fig S5 as shown by the shaded red fill together with red line fit.

The data deviates from the fits that must go through 0,0 at very low surface concentrations close to 0 for the obvious reason that in this region where molecules are almost absent, the statistics are less good.

Supplementary Figure 6) 2D A5 binding curves for the determination of 2D dissociation constants K_d between A5 and higher order oligomers. Plots are arranged into a grid where the columns represent $[A]$ and rows represent the fraction of $[B]$ in complex, $\frac{[AB]}{[AB]+[B]}$. For example, the plot at row 1 column 2 corresponds to the formation of an annexin trimer-of-trimers A_3 from the encounters of A5 and A_2 . Graphs are fitted with eq. 5. 95% confidence interval fits shown by the shaded red fill together with red line fit (data error bars = s.d.).

Minor concerns:

1. The abstract exceeds 150 words extensively.

Yes, currently the abstract was 205 words long.

New abstract 135 words:

Dynamics are fundamental to the functions of biomolecules and can occur on a wide range of time- and length-scales. Here we develop and apply high-speed AFM height spectroscopy (HS-AFM-HS), a technique whereby we monitor the sensing of a HS-AFM tip at a fixed position to directly detect the motions of unlabeled molecules underneath. This gives Angstrom spatial and microsecond temporal resolutions. In conjunction with HS-AFM imaging modes to precisely locate areas of interest, HS-AFM-HS measures simultaneously surface concentrations, diffusion coefficients and oligomer sizes of annexin-V on model membranes to decipher key kinetics allowing us to describe the entire annexin-V membrane-association and self-assembly process in great detail and quantitatively. This work pioneers HS-AFM-HS and displays how it can

assess the dynamics of unlabeled bio-molecules over several orders of magnitude and separate the various dynamic components spatiotemporally.

2. L53: “Whilst able to provide...”, there is no subject in this sentence.

This sentence has been restructured to give the subject:

L45:

X-ray crystallography and electron microscopy (EM), are most powerful techniques to study biomolecular structures,^{4,5} whilst able to provide unparalleled spatial resolution, the structures obtained from these methods are limited by ensemble averaging and static snapshots of fixed conformations.

3. L137: “tip parachuting”, brief explanation is necessary for inexperienced readers, or refer literatures.

We agree with the reviewer and have therefore updated the text as follows:

L129:

The average height and abundance of these streaks across the membrane patch can be used to approximate the surface coverage over time (**Fig. 1c**); however such measurements are prone to error due to tip parachuting (where the tip loses contact with the sample and takes some time to return to the surface) and tip induced movement of the proteins.

4. Fig. 1i: Data at high temporal resolution regime (e.g., 0 – 10 ms) should be also shown to demonstrate the wide variety of height profile detected.

Whilst we agree it is useful to show the high temporal resolution data, high resolution data is already displayed in Fig. S2, Fig. 3 and now Fig. 4 and therefore we feel an additional panel to Fig. 1 is not necessary.

5. Fig. 1d, g: The values beside the images should include the temporal range (e.g., a few ms and ~20 us, respectively) as shown in Fig. 1a.

Values showing the temporal resolution beside images Fig. 1d and g are already shown in the figure.

6. The multiple fitting seen in Fig. 1f is wrong? Although the graph is exactly the same as Fig. 5f bottom, the latter is fitted by the single exponential decay. Or, either is wrong?

In Fig. 1f the line scanning data of rotational dwell times is fit by multiple Gaussians, fitted under the principle that the trimer may have a number of preferred dwell-times depending on its interaction with the surrounding lattice. This was not described in the main text. The main text has therefore been updated to read:

L152:

Analysis of the periods of time spent in each state before rotation (**Fig. 1f**) shows a wide distribution best fit by three Gaussians peaking at 13ms, 41ms and 96ms suggesting possibly three different modes of interaction with the surrounding lattice, that we tentatively assign to the three possible interaction sites of the rotating trimer with its environment.

As described in response to Q12(major) the line in Fig. 5f was a lifetime plot and not a fit. Fig. 5f has been updated to avoid confusion and provide easier direct comparison with height spectroscopy data.

7. Cite a literature for Eq.1. Also, τ_D should be used in the main text.

We agree with the reviewer that a citation for eq. 1 is required and that all terms in the equation should be described in the main text. Also, to avoid confusion with r_p and r_d the text has been updated as follows:

L178:

For proteins undergoing 2D Brownian diffusion, the dwell-time τ_D , of the molecule in a detection area is dependent on the protein's diffusion coefficient (D), and the width of the detection area w , by eq. 1.²⁹

$$\tau_D = w^2/4D \quad (\text{eq. 1})$$

L643:

29. Fahey, P. F. *et al.* Lateral diffusion in planar lipid bilayers. *Science*, **195**, 305-306 (1977).

8. L194: It is valuable to summarize the mean value and estimation error of r_p for each A5 oligomer in a table.

We agree with the reviewer that these values are important for the reader and therefore have added a supporting table to the supplementary information to give details of our calculations:

A5 Oligomer	$\langle d_p \rangle$ (nm)	w (nm)
Monomer	9.4	10.2 ± 0.4
Dimer	13.3	14.2 ± 0.5
Trimer	16.2	17.3 ± 0.6
Tetramer	18.8	20.0 ± 0.6
Pentamer	21.0	22.3 ± 0.7

Table S1. Average oligomer dimensions as determined from the molecular structures of A5 oligomers before ($\langle d_p \rangle$) and after (w) convolution with a 1 ± 0.5 nm AFM tip radius.

Reference to this table has been added to the main text as follows:

L191:

Performing height spectroscopy on A5 molecules undergoing self-assembly into higher order oligomers is therefore expected to produce the multi-peaked distribution of dwell-times we observe in **Fig. 1j**, which not only depends on oligomer size but also its size-dependent diffusion rate (see supporting Table S1 for full details of oligomer dimensions).

9. L202: r is r_p ? Also, explain $t_{z>T}$ and t_{total} in the main text.

Yes, r is r_p . The equation has been updated to reflect this:

$$c = \frac{t_{z>H_T}}{t_{total}} \cdot \frac{1}{r_p^2} \quad (\text{eq. 2})$$

Also, to explain the terms in the equation the following text has been added:

L206:

Where $t_{z>H_T}/t_{total}$ corresponds to the fraction of time the height z , is greater than the threshold height, H_T .

10. L228: It is more kind if the authors add the information of the A5 concentration here.

We agree with the reviewer that including the values in the main text is easier for the reader. The text has been updated as follows:

L234:

As the Ca^{2+} -concentration was increased from $50\mu\text{M}$ to $100\mu\text{M}$, $150\mu\text{M}$ and $200\mu\text{M}$ (Fig. 2c,d,e) we observed increases in both the frequency and dwell-times of events; equating to a three orders of magnitude increase in the surface density of A5 from 1.0 ± 0.6 to 285 ± 150 A5 per μm^2 .

11. Technically, AFM imaging and height spectroscopy cannot be performed at the same time. In this sense, the blue lines from Fig. 4a, b should be drawn outside of the panel of Fig. 4c or removed from the figures.

We agree with the reviewer that this may mislead some readers and have altered Figure 4 accordingly.

12. Typically, the A5 height is detected as 1.0-1.5 nm in HS-AFM-HS. Why the height seen after ~85 s in Fig.4c are so high (more than 2 nm)?

The height after 85s matches well with the thickness of the 2D lattice of annexin. Before formation of a lattice diffusing A5 heights are observed varying between 0.9nm and 2.2nm. Experiments using step functions (as in Fig. S2) show that for dwell times below 90s the height does depend on the dwell time, this can also be observed in much of the experimental data. This effect is due to the z piezo response time, whilst able to respond to a deflection signal, the rate at which z is changed is not always quick enough to reach the full z displacement. To display this effect a panel has been added to supporting figure 2 and the following text has been added to the manuscript:

L168:

A distribution of heights between H_T and ~2nm is observed due the z-feedback not being able to fully respond to the shorter dwell-times (Supplementary Fig. 2c).

13. Fig. 4c: Showing typical three data at high temporal resolution regime (e.g., 20 – 30 ms, 60-70 ms and 80 – 90 ms) are valuable to demonstrate the height profile change over the time and to directly link the results of Fig. d, e.

We agree with the reviewer that such additional panels would be useful to better display the changes over time. Fig. 4 has been updated to reflect this with insets showing the requested high temporal resolution zoom-ins into the HS-AFM-HS trace.

14. k_B and K_b are randomly used. They should be used uniformly.

Instances of K_b have been changed to K_B to create uniform use throughout the manuscript.

15. The blue arrows seen in Fig. 5a should be bidirectional because the A5 rotates both clockwise and counter-clockwise.

Fig. 5a has been updated with bidirectional arrows.

16. L504: "HEPES NaCl" is "HEPES NaOH" correctly?

We thank the reviewer for spotting this typographical error, the sentence should read:

L528:

...buffer solution containing 10mM HEPES at pH 7.4...

Reviewer #2 (Remarks to the Author):

In their manuscript „High-Speed AFM Height Spectroscopy (HS-AFM-HS): Microsecond dynamics of unlabeled biomolecules”, Heath and Scheuring present a methodology for probing diffusional motion of single molecules and their surface concentration at a solid-liquid interface. The authors employ this methodology to quantify the kinetic constants that govern the 2D self-assembly of the protein annexin-V, which is proposed to play a role in harnessing damaged membranes.

The methodology uses a high-speed atomic force microscope that can acquire topographic images of a 2D surface in the ~ 100 ms regime. To improve temporal resolution, the authors reduce the dimensionality of the system. Continuous scanning of a single line, coined “high speed AFM line-scanning”, improves the temporal resolution by two orders of magnitude (~ 1 ms). Further reduction of the dimensionality by probing the height at a single (x,y)-position yields a temporal resolution that is limited by the z-piezo response time, or ~ 20 μ s.

Whereas the concept of reducing the dimensionality to increase the temporal resolution of AFM is not new, this is (to my knowledge) the first example that employs a high-speed atomic force microscope, thereby enabling unprecedented time-resolution. In turn, new applications in biology and possibly in interface science in general might be addressed. While I have some detailed comments that could further improve this work (see below) I find the manuscript well-written and generally interesting.

Detailed comments

1. The term “height spectroscopy” implies that a spectrum of heights can be addressed, and that different heights provide additional information. However, the height information is solely used to address the presence or absence of diffusing proteins under the scanning probe, using a threshold value. How accurate is the measurement of height, and how does this depend on the time it is measured? More specifically, in supplementary figure S2, step-functions with $z = 2$ nm input show decreasing measured heights for reduced dwell times. Can the authors explain this? How does the method perform for different molecular heights, in particular for smaller molecules? It would be insightful to perform experiments wherein the z-piezo is moved to different heights (<2 nm), and for different dwell times, to identify the dependence of the temporal resolution on the height of diffusing molecules.

The reviewer is correct, experiments using step functions (as in Fig. S2c) show that for dwell times below 90μ s the height does depend on the dwell time, this can also be observed in much of the experimental data. This effect is due to the z piezo response time, whilst able to respond to a deflection signal, the rate at which z is changed is not always quick enough to reach the full z displacement. To display this effect a panel has been added to **supporting figure 2** and the following text has been added to the manuscript:

L168:

A distribution of heights between H_T and ~ 2 nm is observed due the z-feedback not being able to fully respond to the shorter dwell times (Supplementary Fig. 2c).

In terms of different heights, we are currently investigating a number of biomolecules with heights including both smaller and larger molecules (~ 1 - 5 nm in height). The HS-AFM-HS method performs well with both larger and smaller molecules, the results of these experiments will however form the subjects of upcoming manuscripts.

In the present work, the HS-AFM-HS data on the rotating A5 (Fig. 5c, 5f top) give readily a glimpse of these capabilities. The molecule is always present under the tip and changes height (due to rotation) by only about 0.3nm, accurately detectable.

2. The method quantifies dwell times and frequencies of molecules moving under the tip. Other parameters are determined indirectly and are subject to error propagation or fitting quality. The authors should provide appropriate statistical analysis (reduced chi-square of fits, confidence intervals for the fit parameters, ...) and indicate the meaning of the errors on values where appropriate.

We agree with the reviewer and errors values have now been added throughout the manuscript. Standard deviation values for the dwell time fittings in Fig. 2 and surface concentrations have been added to the main text. Also, the propagated error values have been added to the D values discussed in the main text. Errors values based on the 95% confidence intervals of the fitting of equation 5 to the data have also been added to the K_D values in table 1 and fits in Fig. S5. Meanings of errors have also been added where an error of a value is mentioned for the first time in the text.

3. Supported lipid bilayers as a substrate are less dynamic than “free-standing” bilayers. The authors should comment on how this would affect the dynamics of annexin-V assembly, and to what extent the kinetics can be translated to a cellular context.

We agree with the reviewer that supported bilayers are less dynamic than free-standing model bilayers. However, in a cellular context the situation is again very different, and number of diffusion barriers and variations in lipid composition mean that dynamics of cellular membranes have diffusion behavior that matches more closely to that of supported bilayers.

Furthermore, the system studied here, diffusion of A5, a membrane-associated protein only peripherally attached to the upper leaflet of the supported bilayer, is likely not that much influenced by the support.

4. Drift correction is performed using a “laboratory built image analysis software”. How does this software deal with drift, and (how) is it applied to HS-AFM-LS and HS-AFM-HS?

Drift of HS-AFM images are corrected using image correlation alignment software to find the x-y translations required to align the image set (this is a post data capture method). Although not used here, algorithms can be applied to correct for drift in the fast scan direction (x) of HS-AFM-LS data using the z, x values. HS-AFM-HS data is and can not be drift corrected, drift can however be minimized and estimated by pre- and post- HS-AFM-HS imaging.

To assess the drift, we use HS-AFM images of the annexin lattice for several minutes and then use the image correlation alignment software to find the x-y translations required to align the image set. Under normal imaging conditions the total drift distance was found to vary from as low as 0.02 nm/s with a well-equilibrated system (as shown in supplementary Fig. 8) up to 0.1nm/s. This is drift relatively slow 10-50s/nm in comparison to the tip radius and the area of interest, which is an additional advantage of HS-AFM-HS over fluorescence techniques where bleaching often limits the total time a molecule can be observed with high temporal resolution.

Furthermore, HS-AFM-HS is used when monitoring fast dynamic events beyond the speed of acquisition in imaging mode (much faster than 10s⁻¹), thus even for the cases when the drift is relatively strong, one still captures hundreds of events on 1nm within 10s. If one is to analyze slower dynamics, then line scanning or imaging mode is appropriate and drift can be corrected based on the recorded environmental features in either one dimension (line scanning) or two dimensions (imaging).

To minimize x-y drift, the scanner and tip holder were placed in stable positions and large x-y translations were avoided directly before HS-AFM-HS capture. Additionally, to ensure the AFM tip is on the same molecule for a certain period, HS-AFM image sets were taken directly before and after HS-AFM-HS measurements were captured, allowing the total drift to be measured.

The following supplementary figure and movie S2 have been added to the supporting information:

Supplementary Figure 8. Quantification of lateral drift. a) x-y position change during HS-AFM imaging as determined by image correlation alignment with subpixel interpolation of supporting movie S2. The translations are overlaid onto an annexin trimer (A5) to illustrate the scale of lateral drift during HS-AFM-HS measurements. b) Total drift distance from initial position over time with linear fit to determine a drift rate of 51s/nm (0.0195 nm/s). It should be noted that alignment was performed on images with 0.5nm/pix and thus apparent short timescale drift noise is expected to be due to alignment accuracy.

The following text has also been added to the main text of the manuscript:

L394:

It should be noted the scanner stage can drift by some nanometers in x- and y-dimension, especially during the seconds after execution of a position or scan range change due to piezo-relaxation. To assess the mechanical drift, we can capture HS-AFM images of the annexin lattice for several minutes and then use image correlation alignment software to find the x-y translations required to align the image set. Under normal imaging conditions the total drift distance was found to vary from as low as 0.02 nm/s with a well-equilibrated system (as shown in supplementary Fig. 8) up to 0.1nm/s. This is drift relatively slow 10-50s/nm in comparison to the tip radius and the area of interest and suggests HS-AFM-HS can be positionally accurate for 10s of seconds. Such stability provides an additional advantage of HS-AFM-HS and HS-AFM-LS over fluorescence techniques where bleaching often limits the total time a molecule can be observed with high temporal resolution.

Added to methods:

L552:

To minimize x-y drift, the scanner and tip holder were made sure to in stable positions and large x-y translations were avoided directly before HS-AFM-HS capture. Additionally, to ensure the AFM tip is on the same molecule for a certain period and no significant changes in tip radius occur, HS-AFM image sets were taken directly before and after HS-AFM-HS measurements were captured, allowing the total drift to be measured and tip quality to be assessed.

5. Tip size can change during the measurement. How does tip size affect the measured diffusion times of annexin-V?

Indeed, tip size can change during measurement. An increase in tip size increases the area in which a molecule can be detected and thus increases the overall dwell time, for example an increase in tip radius from 1 to 2 nm would increase w by ~ 2 nm and thus the dwell of an A5 molecule from $33\mu\text{s}$ to $47\mu\text{s}$. This variation is within the standard deviation of the dwell times measured.

Since we are able to asses tip quality before and after each height spectroscopy data capture (typically 60s) through HS-AFM imaging and we have not observed significant overall shifts in dwell time during measurements over long periods we don't believe any significant changes in tip radius typically occur during our height spectroscopy measurements.

A comment on this has be added to the methods:

L555:

Additionally, to ensure the HS-AFM tip is on the same molecule for a certain period and no significant changes in tip radius occur, HS-AFM image sets were taken directly before and after HS-AFM-HS measurements were captured, allowing the total drift to be measured and tip quality to be assessed.

Reviewer #3 (Remarks to the Author):

The authors developed high-speed AFM height spectroscopy for directly detecting the motions of molecules under a HS-AFM tip at a fixed position. They measured simultaneously surface concentrations, diffusion coefficients and oligomer sizes of annexin-V on model membranes and described the entire annexin-V membrane-association and self-assembly process in great detail and quantitatively by using HS-AFM-HS and HS-AFM imaging. Their method has Angstrom spatial and microsecond temporal resolutions of unlabeled molecular fluctuations. In my opinion, this work is interesting and could be published in Nature Communication. However, before publication, I would like to propose a revision on the following points.

1. Did different ratios of DOPC:DOPS affect the annexin-V membrane-association and self-assembly process?

The reviewer makes an interesting point. We did not vary the ratios of DOPC:DOPS in this study, instead we chose to investigate a typical inner membrane composition of 80:20 PC/PS. However different ratios have been previously studied with other techniques, at low and medium PS content (5–20%) A5 binding is less and forms the open hexagonal arrays with $p6$ symmetry that we observe in our study, while at high PS-content ($\geq 40\%$) crystals with $p3$ symmetry dominated, in these studies. Findings of the different PS concentrations are well summarized in reference 25 and 28. These previous studies were not able to assess the self-assembly process

and we agree it would be interesting to observe how 2D association kinetics vary with varying PS concentrations.

2. The authors determined the diffusion coefficient of dimers and trimer of A5. Can they estimate their theoretical values? Can they compare their results with other methods?

As shown in Fig. S4, two different fits to the data show the theoretical decrease in diffusion constant based on two different models, the Saffman-Delbrück model and Stokes-Einstein model with data showing the best fit to the Saffman-Delbrück model as described in the text:

Diffusion constants, derived from the dwell-time peaks, show a decrease with increasing oligomer size, consistent with the Saffman-Delbrück approximation.³²

Comparison of the A5 trimer diffusion coefficient is made with data obtained by FRAP, line 225:

A5 diffusion has previously been shown to be of the order $1 \mu\text{m}^2/\text{s}$ using FRAP,³⁰ in good agreement with the $0.77 \mu\text{m}^2/\text{s}$ found here.

However, no data exists for the A5 dimer and higher order oligomers thus no such comparison can be made for these states.

3. The authors mentioned that “the A5 p6-lattice assembled, it is made up of two-thirds p6-trimers that constitute the ‘honeycomb lattice’ and one-third non-p6 trimers that are not strictly part of the lattice being trimers sitting on the 6-fold symmetry axis”. Why?

The A5 lattice has a non-crystallographic trimer located in the p6 axis interstice, this trimer, unlike the fixed trimers in the honeycomb p6 lattice, has a high degree of rotational freedom. Additionally, this trimer that resides at the 6-fold symmetry axis can bind and unbind to the membrane in a calcium dependent manner without compromising the p6 lattice as shown in our previous work (reference 27). The biological function of this trimer is unclear but it may facilitate easier assembly of the p6 lattice whilst providing freedom to assemble a lattice around other membrane proteins.

We mention this because it is this trimer’s rotation, which we investigate to showcase and compare the capabilities of line scanning and height spectroscopy.

4. Can this method be used for monitoring the annexin-V membrane-association and self-assembly process on the surface of cell?

Although this would be very interesting, technically it is currently not possible because the *in vivo* annexin-V membrane assembly process occurs on the inside of cells it would not be possible for the AFM tip to be able to access the inside of the cell without damaging the cell membrane. It would however be possible to look at other processes on the outer cell membrane or the inner membrane by using unroofed cells.

REVIEWERS' COMMENTS:

Reviewer #1 (Remarks to the Author):

The reviewer greatly thanks that the authors responded to my concerns. Most of them were well address and the necessary changes were made, which dramatically improved the manuscript. However, the reviewer points out the followings that are still unclear.

Major concerns:

M1, M8

The reviewer agreed with the authors' thought about the current title.

Concerning the temporal resolution of HS-AFM-HS, the reviewer did not agree with that, based on Supplementary Figure 2, the authors' statement that ~20 us is a rather conservative estimate.

However, this mismatch between the reviewer and the authors should result from that the authors did not show any height/time traces of A5 showing pulses of less than 10 us in the manuscript. Judging from the dwell time plots, those pulses were actually obtained. The authors should just show such height/time traces elsewhere, by which all reader will understand the authors' statement.

Also, the reviewer agrees that the deflection or amplitude signal will give better temporal resolution. The reviewer is looking forward to a new article by the authors using those data.

M6

To draw the line of H_T onto the all height/time traces dose not clutter the figures. Drawing the line of H_T makes very clear what the method is doing and gains the readability too. The reviewer did not understand why the authors hardly refuse this. Is there any inconvenience?

M11

In the response letter, the authors said that the values were obtained from 6 different trimers. However, this number is 3 in the text. Which is correct?

M13

The authors state that there are similarity between the data and model. However, there are clear discrepancies between the data and model. For example, see the left side on the "Counterclockwise" or the right side on the "Oscillate". Current way of writing will mislead the readers. The authors should clearly explain that the model was not created by fitting data.

Minor concerns:

m13

Additional panels are very nice. But, the authors should explain the color difference of lines.

Reviewer #2 (Remarks to the Author):

In their response and in their revised manuscript, the authors have properly addressed the comments and suggestions of the referees. I believe the manuscript is almost ready for publication, but have two minor comments/ suggestions.

1. P4, lines 152-156. The authors should substantiate their claim "the distribution is best fit by the sum of three Gaussian" by providing the reduced chi-square value of the fit, and by comparing this value with reduced chi-square values for fits to different numbers of Gaussians.

2. It would be useful to explicitly state that A5 is only peripherally attached to the upper leaflet of the supported bilayer

Reviewer #1 (Remarks to the Author):

The reviewer greatly thanks that the authors responded to my concerns. Most of them were well address and the necessary changes were made, which dramatically improved the manuscript. However, the reviewer points out the followings that are still unclear.

Major concerns:

M1, M8

The reviewer agreed with the authors' thought about the current title.

Concerning the temporal resolution of HS-AFM-HS, the reviewer did not agree with that, based on Supplementary Figure 2, the authors' statement that ~20 us is a rather conservative estimate. However, this mismatch between the reviewer and the authors should result from that the authors did not show any height/time traces of A5 showing pulses of less than 10 us in the manuscript. Judging from the dwell time plots, those pulses were actually obtained. The authors should just show such height/time traces elsewhere, by which all reader will understand the authors' statement.

We agree with the reviewer that showing 10us events will allow the reader to understand our statement of having 10us time resolution and have therefore updated Figure 3b to show an example of a 9us pulse.

Also, the reviewer agrees that the deflection or amplitude signal will give better temporal resolution. The reviewer is looking forward to a new article by the authors using those data.

M6

To draw the line of H_T onto the all height/time traces dose not clutter the figures. Drawing the line of H_T makes very clear what the method is doing and gains the readability too. The reviewer did not understand why the authors hardly refuse this. Is there any inconvenience?

We agree with the reviewer that such lines may help the reader and have updated Fig. 2 with dashed lines behind the raw data to show H_T.

M11

In the response letter, the authors said that the values were obtained from 6 different trimers. However, this number is 3 in the text. Which is correct?

Reliable transitions were obtained from 3 different trimers by line scanning and 3 different trimers for height spectroscopy. In the response letter we refer to the total number by both techniques (i.e. 6) however in the manuscript we refer to 3 different trimers measured by each technique.

In the caption of Fig. 5 we state:

"Histograms each contain data from 3 different trimers each showing no significant statistical differences between molecules."

We feel this is clear for the reader.

M13

The authors state that there are similarity between the data and model. However, there are clear discrepancies between the data and model. For example, see the left side on the "Counterclockwise" or the right side on the "Oscillate". Current way of writing will mislead the readers. The authors should clearly explain that the model was not created by fitting data.

We agree with the reviewer that some readers may believe the model is fitted to the data, therefore we have updated the text to explicitly state the model rotations are not fitted to the data:

"As visible in raw data (Fig. 5e, top) and corroborated by model line scanning rotations (not fitted to the data) (Fig. 5e, bottom), the characteristics of these transitions depend on the initial state and the direction of rotation (Supplementary Fig. 1)."

Minor concerns:

m13

Additional panels are very nice. But, the authors should explain the color difference of lines.

The additional panels (Fig. 4b) show a number of example diffusion event peaks from the annotated time regions, the colour difference is only used to allow the reader to distinguish each trace belonging to each peak.

To clarify for the reader the figure caption has been updated as follows:

Fig.4:...

d), e) and f) show higher temporal resolution zoom-ins of the HS-AFM-HS trace (b) showing example diffusion events from the 20-30s, 60-70s and 80-87s time regions, respectively, with different line colors representing different events.

Reviewer #2 (Remarks to the Author):

In their response and in their revised manuscript, the authors have properly addressed the comments and suggestions of the referees. I believe the manuscript is almost ready for publication, but have two minor comments/suggestions.

1. P4, lines 152-156. The authors should substantiate their claim “the distribution is best fit by the sum of three Gaussian” by providing the reduced chi-square value of the fit, and by comparing this value with reduced chi-square values for fits to different numbers of Gaussians.

We agree with the reviewer that discussing the reduced chi squared values of the fits to different numbers Gaussians would strengthen the argument that the line scanning data is best fit by three gaussians.

The reduced Chi-square values for fitting to different numbers of Gaussians are as follows:

1 Gaussian fit = 25.6
2 Gaussian fit = 2.95
3 Gaussian fit = 1.23
4 Gaussian fit = 0.6

Suggesting that a three Gaussian fit is the most likely fit.

To convey this to the reader the text has been updated:

“Analysis of the periods of time spent in each state before rotation (Fig. 1f) shows a wide distribution best fit by three Gaussians (as determined by reduced chi-squared values) peaking at 13ms, 41ms and 96ms suggesting possibly three different modes of interaction with the surrounding lattice, that we tentatively assign to the three possible interaction sites of the rotating trimer with its environment.”

2. It would be useful to explicitly state that A5 is only peripherally attached to the upper leaflet of the supported bilayer.

We agree with the reviewer that it is important for the reader to understand that annexin binds to the surface of the membrane and is not embedded in the lipid bilayer. We have therefore updated the text as follows:

HS-AFM imaging of supported lipid bilayers (SLBs) containing 20% phosphatidylserine (Fig. 1a) shows how annexin binding to the surface of the membrane (upper leaflet) and subsequent self-assembly occurs over the second timescale (Fig. 1b).